# Cell types and neuronal circuitry underlying female aggression in *Drosophila*

Catherine E Schretter[1], Yoshinori Aso[1], Alice A Robie[1], Marisa Dreher[1], Michael-John Dolan[1,2], Nan Chen[1], Masayoshi Ito[1], Tansy Yang[1], Ruchi Parekh[1], Kristin M Branson[1], Gerald M Rubin[1]*

[1]Janelia Research Campus, Howard Hughes Medical Institute, Ashburn, United States; [2]Current address: Department of Neurology and F.M. Kirby Neurobiology Center, Boston Children's Hospital, Harvard Medical School, Boston, United States

**Abstract** Aggressive social interactions are used to compete for limited resources and are regulated by complex sensory cues and the organism's internal state. While both sexes exhibit aggression, its neuronal underpinnings are understudied in females. Here, we identify a population of sexually dimorphic aIPg neurons in the adult *Drosophila melanogaster* central brain whose optogenetic activation increased, and genetic inactivation reduced, female aggression. Analysis of GAL4 lines identified in an unbiased screen for increased female chasing behavior revealed the involvement of another sexually dimorphic neuron, pC1d, and implicated aIPg and pC1d neurons as core nodes regulating female aggression. Connectomic analysis demonstrated that aIPg neurons and pC1d are interconnected and suggest that aIPg neurons may exert part of their effect by gating the flow of visual information to descending neurons. Our work reveals important regulatory components of the neuronal circuitry that underlies female aggressive social interactions and provides tools for their manipulation.

*For correspondence:
rubing@janelia.hhmi.org

Competing interests: The authors declare that no competing interests exist.

## Introduction

Aggressive behaviors are important for gaining access to resources, including food and territory, and are exhibited by both sexes in multiple species (*Anderson, 2016*; *Kravitz and Huber, 2003*; *Zwarts et al., 2012*). As aggressive actions carry the risk of injury, strict regulation of aggression is needed to facilitate survival. Sensory information about the presence of other individuals and the nature of the surrounding environment strongly modulate aggressive social interactions (*Chen and Hong, 2018*; *Hoopfer, 2016*). However, understanding the neuronal mechanisms by which such stimuli influence aggression has been hindered by a lack of knowledge about the structure of the underlying neuronal circuits, particularly in females.

Centers mediating, or conveying the information necessary for, aggression have been identified in the medial hypothalamus through classic experiments using electrical stimulation in cats and rodents (*Albert et al., 1979*; *Bandler et al., 1972*; *Berntson, 1973*; *Chi and Flynn, 1971*; *Gregg, 2003*; *Kruk et al., 1983*; *Lammers et al., 1988*; *Siegel et al., 1999*; *Takahashi and Miczek, 2014*; *Woodworth, 1971*). Such key regions are thought to perform a different role than other brain areas that facilitate aggressive interactions by altering the overall level of social behavior (*Siegel et al., 1999*). Recent work using opto- and chemo-genetic techniques have narrowed down these key regions to small populations of cells in mice, including those expressing estrogen receptor alpha (Esr1) and progesterone receptor (PR) in the ventrolateral part of the ventromedial hypothalamus (VMHvl) (*Hashikawa et al., 2017*; *Lee et al., 2014*; *Yang et al., 2013*). While Esr1[+] neurons in the VMHvl regulate aggression in both male and female mice, there are sex differences in the

populations involved (*Hashikawa et al., 2017*). Additionally, the VMHvl has been implicated in other female sexual behaviors (*Hashikawa et al., 2017*; *Lee et al., 2014*; *Pfaff and Sakuma, 1979a*; *Pfaff and Sakuma, 1979b*; *Yang et al., 2013*), further complicating the identification of the specific cell types that mediate aggressive interactions.

Since the first observation of aggressive behaviors in *Drosophila* by Sturtevant in 1915, social behaviors associated with attack and threat displays in flies have been well described ethologically (*Shelly, 1999*; *Sturtevant, 1915*; *Ueda and Kidokoro, 2002*; *Zwarts et al., 2012*). While male aggression is heightened in the presence of mate-related cues, female flies display increased aggressive behaviors when nutrients are limited and near egg laying sites (*Bath et al., 2017*; *Bath et al., 2018*; *Lim et al., 2014*; *Shelly, 1999*; *Ueda and Kidokoro, 2002*). Additionally, social isolation can increase aggression in both male and female flies (*Hoffmann, 1990*; *Ueda and Kidokoro, 2002*). As in mammals (*Hashikawa et al., 2017*), aggressive behaviors in flies include sex-specific components, such as head butting in females, as well as those that are shared between the sexes (*Nilsen et al., 2004*). Due to the complexity of the behavior and the sensory stimuli that influence its presentation, a circuit diagram would greatly facilitate understanding the underlying neuronal mechanisms. To gain a mechanistic understanding of how these behaviors are regulated and executed, we will need to identify the specific cells that contribute in each sex and place them in the context of larger neuronal circuits.

*Drosophila melanogaster* provides a good model for dissecting the neuronal circuitry of aggression due to the genetic tools available for targeting and manipulating individual cell types, the availability of extensive connectomic information, and the relative simplicity of its nervous system and behavior (*Bellen et al., 2010*; *Dionne et al., 2018*; *Kravitz and Huber, 2003*; *Scheffer et al., 2020*; *Simpson and Looger, 2018*; *Tirian and Dickson, 2017*). In male flies, studies investigating the neuronal correlates of aggression have implicated a group of 18–34 cells in the central brain, the P1/ pC1 cluster, as well as various neuropeptides and biogenic amines, including neuropeptide F, tachykinin, and octopamine (*Alekseyenko et al., 2019*; *Alekseyenko et al., 2014*; *Asahina, 2018*; *Asahina, 2017*; *Asahina et al., 2014*; *Dierick and Greenspan, 2007*; *Hoopfer et al., 2015*; *Hoyer et al., 2008*; *Ishii et al., 2020*; *Wohl et al., 2020*; *Wu et al., 2020*; *Zhou et al., 2008*). However, research on female aggressive social interactions has been less extensive as females exhibit less aggression under the same behavioral conditions used for males. There are also sex differences in the behavioral components and underlying neurons important for aggression (*Hoopfer et al., 2015*; *Nilsen et al., 2004*). Genes involved in sexual differentiation, including *doublesex* (*dsx*) and *fruitless* (*fru*), contribute to social behaviors (*Dickson, 2008*; *Koganezawa et al., 2016*; *Pavlou and Goodwin, 2013*; *Siwicki and Kravitz, 2009*; *Yamamoto, 2007*; *Yamamoto and Koganezawa, 2013*; *Vrontou et al., 2006*; *Zhou et al., 2014*). Recent work has revealed the involvement of the *dsx*-expressing pC1 cluster, a group of 5 cell types, in promoting aggressive phenotypes in female flies (*Deutsch et al., 2020*; *Fathy, 2016*; *Palavicino-Maggio et al., 2019*). As in the VMHvl of mice, cells within this cluster can be divided into multiple subtypes and particular subtypes are also involved in other female behaviors, including mating and egg laying (*Wang et al., 2020a*; *Wang et al., 2020b*). Understanding the flow of information within the neuronal circuit controlling aggression will require knowledge of which cells within the pC1 cluster contribute to aggressive behaviors.

We used connectomic, genetic and behavioral analyses to characterize neuronal cell types contributing to female aggressive behaviors in *Drosophila*. We found that optogenetic activation of a subset of the neurons derived from the aIP-g neuroblast (*Cachero et al., 2010*) increased female aggression, even in the absence of aggression-promoting environmental conditions. Importantly, blocking synaptic transmission from these neurons resulted in diminished female aggression, indicating that these cells normally play a role in modulating social interactions. We next identified a specific, single cell type within the pC1 cluster (pC1d) that induces aggression upon activation. Analysis of the connectome of a large part of the fly central brain (*Scheffer et al., 2020*) revealed that aIPg and pC1d neurons are strongly interconnected as well as uncovered multiple other neurons linked to these two cell types. Our anatomical data did not allow us to determine which of these neurons play a role in aggressive behaviors, but they did provide clues. For example, we uncovered a simple circuit motif by which aIPg activation could increase the saliency of visual information from LC10 neurons, a cell type used by males to track potential mates during courtship (*Ribeiro et al., 2018*; *Sten et al., 2020*). We also identified several short neuronal paths connecting outputs of aIPg

neurons to descending interneurons that drive motor behavior. Taken together, our work yields insights into female aggressive behavior and identifies two cell types that appear to form key nodes of the circuit underlying aggression.

## Results

### Identification of neurons involved in female aggressive behaviors

In a behavioral screen using split-GAL4 lines to examine another phenotype, we noted a dramatic increase in female social interactions, including known components of aggression (*Bath et al., 2017*; *Nilsen et al., 2004*; *Palavicino-Maggio et al., 2019*; *Ueda and Kidokoro, 2002*), upon stimulation of a neuronal subset of approximately eleven cells (*Figure 1A,B*). We generated multiple, independent split-GAL4 lines (*Dionne et al., 2018*; *Luan et al., 2006*; *Pfeiffer et al., 2010*) labelling this same neuronal population. Group-housed virgin females from these lines that also expressed the red-shifted opsin CsChrimson were then screened for behavioral changes upon light activation (*Kim et al., 2015*; *Klapoetke et al., 2014*). We performed automated behavioral analyses using video-assisted tracking software to monitor freely moving flies within a 127 mm arena (*Branson et al., 2009*; *Robie et al., 2017*; *Simon and Dickinson, 2010*). Upon stimulation, the additional split-GAL4 lines labelling this neuronal population (*Figure 1—figure supplements 1* and *2*) exhibited similar increases in social interactions that included aggressive behaviors (*Figure 1D–G*, *Figure 1—figure supplement 3*; compare *Videos 1* and *2*).

To characterize these cells, we used whole-mount immunohistochemistry directed against the same construct used for optogenetics, mVenus-tagged CsChrimson. The split-GAL4 lines labelled a set of neurons with cell bodies located in the inferior protocerebrum and major projections in the anterior optic tubercle (AOTU), anterior ventrolateral protocerebrum (AVLP), superior medial protocerebrum (SMP), and superior intermediate protocerebrum (SIP) (*Figure 1A,B*, *Figure 1—figure supplements 1* and *2*). The cells observed in our split-GAL4 lines morphologically resemble, and appear to be a subset of, the 32 neurons in the aIP-g neuroblast clone described in *Cachero et al., 2010*. These neurons were previously classified as *fru*[+] auditory interneurons with sexually dimorphic projections (*Cachero et al., 2010*). We refer to the subset we identified as aIPg neurons. In two of our aIPg lines (aIPgSS1 and aIPgSS4), no expression was seen in males (*Figure 1C*, *Figure 1—figure supplement 1C*). We tested for changes in male behavior by optogenetically stimulating males from the aIPgSS1 line and found no differences (*Figure 1—figure supplement 4*). RNAseq analysis of multiple pooled aIPg neurons, purified based on expression in our split-GAL4 lines, confirmed the expression of *fru* and the use of acetylcholine as a neurotransmitter, consistent with previous descriptions (*Cachero et al., 2010*), as well as revealed expression of short neuropeptide F (*sNPF*) and its receptor (*Figure 1—figure supplement 5*).

### Activation of aIPg neurons evokes aggressive behaviors in females

Social behaviors including aggression are comprised of multiple behavioral components and patterns (*Nilsen et al., 2004*). We began by quantifying two such behaviors, chasing and touching, using a set of previously created and validated automatic behavior classifiers (*Robie et al., 2017*). We found that flies increased touching compared to the empty split-GAL4 control during a 30 s stimulation (*Figure 1—figure supplement 6A,B*). A low level of chasing was also detected upon activation (*Figure 1—figure supplement 6C,D*), consistent with the behavioral pattern described for female aggression (*Nilsen et al., 2004*). As a control, we also examined walking behavior. We did find a sharp decrease in the percent of flies walking following stimulus onset, coincident with the increase in touching, but the average walking velocity after stimulation did not differ from that of controls (*Figure 1—figure supplement 6A,E,F*). Examination of behavior metrics for individual flies revealed a significant increase in the number of flies within two body lengths during stimulation (*Figure 1—figure supplement 6G*), consistent with aIPg activation increasing the likelihood of engaging in social behaviors.

Touching and chasing are also components of social interactions, including aggression, in male flies (*McKellar et al., 2019*; *Nilsen et al., 2004*). There are, however, many sex-specific aspects of aggression, including head butting in females and the way in which behavioral patterns progress during an encounter (*Nilsen et al., 2004*). To examine female-specific attributes, we generated and

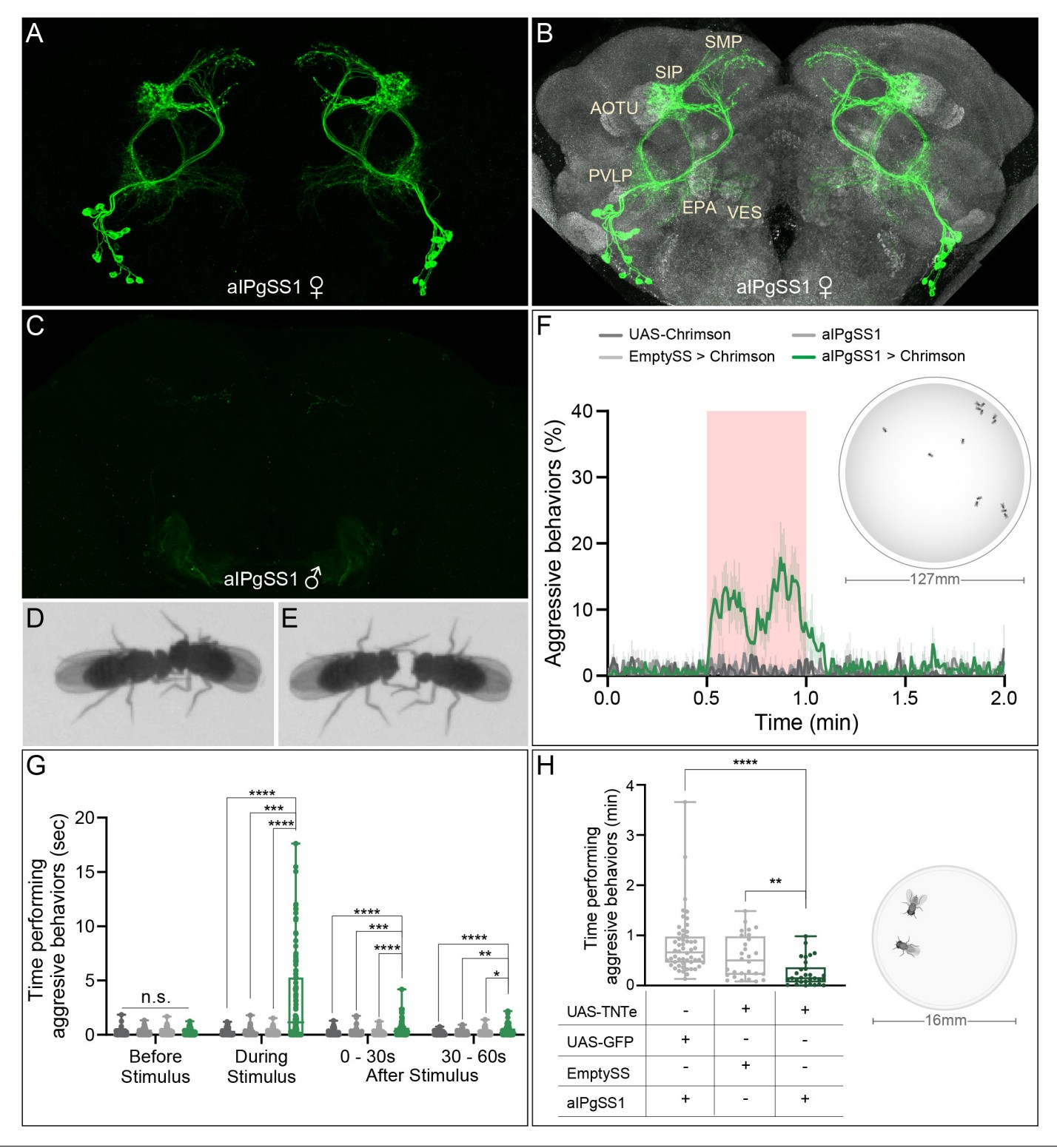

**Figure 1.** Activation of 11 aIPg neurons increases, while inactivation decreases, female aggressive behaviors. (**A, B**) Maximum intensity projection (MIP; 63x) image of the central brain of a female from the aIPgSS1 line crossed with 20xUAS-CsChrimson::mVenus and stained with anti-GFP antibody. Major neuropils innervated are indicated in (**B**) along with the reference stain (nc82) in gray. (**C**) A male brain of the same genotype (MIP; 20x) without the reference stain. Images of the complete brain and ventral nerve cord of a female and male are shown in *Figure 1—figure supplement 1A–C*. Images of individual aIPg neurons, generated by stochastic labeling are shown in *Figure 1—figure supplement 2*. (**D,E**) Images of female flies displaying head butting (**D**) and fencing (**E**) behaviors. (**F**) Percentage of flies engaging in aggressive behaviors over the course of a 2 min trial during which a 30 s 0.4

*Figure 1 continued on next page*

*Figure 1 continued*

mW/mm$^2$ continuous light stimulus (pink shading) was delivered. The mean is represented as a solid line and shaded bars represent standard error between experiments. The timeseries shows the percentage of flies performing aggression displayed as the mean of 0.5 s (15-frame) bins. See *Figure 1—figure supplement 6H* for per experiment quantification. See *Supplementary file 1* and methods for a description of the JAABA classifier. Data was pooled from two independent biological replicates, which included separate parental crosses and were collected on different days. Inset image shows the arena size for F and G, for more detailed view see *Figure 1—figure supplement 3B,C*. (G) Total time an individual spent performing aggressive behaviors during each of four 30 s periods: prior to, during, immediately following, and 30–60 s after the stimulus. Individuals were pooled over two independent testing days from two separate parental crosses. Points represent individual flies. (H) Total time an individual spent performing aggressive behaviors over a 30 min trial. Individuals were pooled over two independent testing days during the same week that were from the same parental cross. Inset image shows arena size, for more details see *Figure 1—figure supplement 10E* and Methods. Points indicate individual flies. Data supporting the plots shown in panels F-H were as follows: F: 20xUAS-CsChrimson, n = 5 experiments; aIPgSS1, n = 5 experiments; EmptySS > 20xUAS-CsChrimson, n = 5 experiments; aIPgSS1 > 20xUAS-CsChrimson, n = 7 experiments. G: 20xUAS-CsChrimson, n = 71 flies; aIPgSS1, n = 65 flies; EmptySS > 20xUAS-CsChrimson, n = 78 flies; aIPgSS1 > 20xUAS-CsChrimson, n = 100 flies. H: aIPgSS1 > UAS-GFP, n = 54 flies; EmptySS > UAS-TNTe, n = 28 flies; aIPgSS1 > UAS-TNTe, n = 30 flies. Data are representative of at least two independent biological repeats, one of which is shown here; see *Supplementary file 3* for exact p-values for each figure. For biological repeats of H, see *Figure 1—figure supplement 11*. Box-and-whisker plots show median and IQR; whiskers show range. A Kruskal-Wallis and Dunn's post hoc test (G, H) was used for statistical analysis on each time point tested. Asterisk indicates significance from 0: *p<0.05; **p<0.01; ***p<0.001; ****p<0.0001.

The online version of this article includes the following source data and figure supplement(s) for figure 1:

**Source data 1.** Source data for behavioural experiments in *Figure 1*.
**Figure supplement 1.** Expression patterns of aIPg split-GAL4 lines.
**Figure supplement 2.** Morphologies of individual aIPg1 - 3 neurons.
**Figure supplement 3.** Additional aIPg split-GAL4 lines also induce aggressive behavior.
**Figure supplement 4.** Optogenetic stimulation of aIPgSS1 > Chrimson males does not result in aggressive behavior.
**Figure supplement 5.** aIPg neurons are cholinergic, fru$^+$ and sNPF$^+$; pC1 neurons are cholinergic and dsx$^+$.
**Figure supplement 6.** Changes in behavioral metrics in females following activation of aIPgSS1 neurons.
**Figure supplement 7.** Optogenetic activation of aggression with Chrimson requires feeding all *trans*-retinal.
**Figure supplement 8.** Behavioral effects of effector strength and stimulus delivery.
**Figure supplement 9.** Higher frequency optogenetic stimulation increases the persistence of aggressive behaviors.
**Figure supplement 10.** Inactivation of aIPg neurons using additional split-GAL4 lines also decreases aggressive behaviors.
**Figure supplement 11.** aIPg inactivation reproducibly decreases aggressive behaviors but not velocity.

validated a new JAABA classifier for female aggression (*Supplementary file 1*). Female aggression encompasses a range of behaviors involved in attack and threat displays; however, in this paper we used the term 'aggression' in a limited way to refer to shoving, fencing, and head butting behaviors. As these behaviors were not always distinguishable at the image resolution used for quantification, an aggressive event was defined as either an instance of head butting (*Figure 1D*) and/or fencing (*Figure 1E*). Similarly, high posture fencing behavior was not distinguished from shoving as we were not able to clearly discern leg posture (*Zwarts et al., 2012*). Examples of fencing and head butting are shown at high spatial and temporal resolution in *Video 3*. Employing this classifier, we found that a 30 s stimulation of the split-GAL4 line aIPgSS1 increased the percentage of flies engaged in

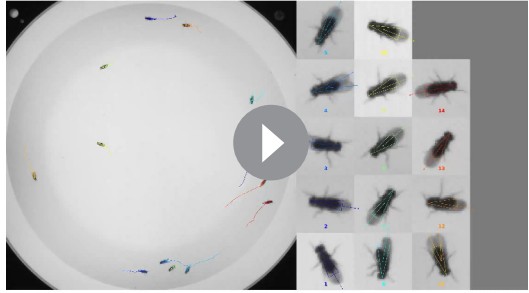

**Video 1.** Behavior of EmptySS > Chrimson flies.
https://elifesciences.org/articles/58942#video1

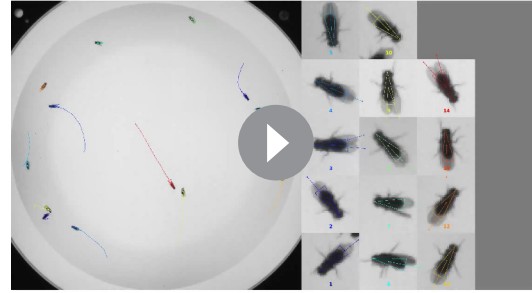

**Video 2.** Behavior of aIPgSS1 > Chrimson flies.
https://elifesciences.org/articles/58942#video2

aggressive behaviors as well as the amount of time individuals spent in such interactions (*Figure 1F,G*, *Figure 1—figure supplement 6H*, compare *Videos 1* and *2*). While only about 15% of the flies are engaged in aggressive behaviors at a given time, over 60% of the flies performed aggressive behaviors at some point during the 30 s stimulation (*Figure 1F,G*). Three different split-GAL4 lines labelling this neuronal population (aIPgSS1, aIPgSS2 and aIPgSS3; *Figure 1—figure supplement 1*) exhibited similar increases in aggressive behaviors upon stimulation (*Figure 1—figure supplement 3*). Consistent with the behaviors being optogenetically induced, these interactions were virtually absent when all *trans*-retinal was omitted from the food (*Figure 1—figure supplement 7*). Activated female flies continued to perform these behaviors at levels higher than control flies for tens of seconds post-stimulation (*Figure 1F,G*, *Figure 1—figure supplement 3*). In sum, using previously validated and new classifiers, we found that aIPg activation results in both general and female-specific components of aggression.

Both the expression level of the effector and the light intensity used for optogenetic stimulation can influence behavior and, in extreme cases, be cytotoxic (unpublished observations; *Kim et al., 2015*). Higher levels of stimulation also increase the possibility that cells expressing the effector at levels too low for detection may contribute to the observed behavior. For these reasons, we examined the expression patterns of our split-GAL4 lines with the highest level of the effector used in our behavioral experiments. We did not detect expression in other cell types or obvious toxicity in the aIPg cells (*Figure 1*, *Figure 1—figure supplement 1*). We also conducted experiments using 5xUAS-, 10xUAS-, and 20xUAS-CsChrimson constructs that are expected to produce a four-fold range of effector expression (*Pfeiffer et al., 2010*; *Figure 1—figure supplement 8A*). Finally, we varied light intensity over a 10-fold range, 0.04, 0.1 and 0.4 mW/mm$^2$ (*Figure 1—figure supplement 8B,C*). Similar effects on aggressive behavior were found under all conditions, strongly supporting the conclusion that the cell types we observe by confocal imaging are the ones responsible for mediating the observed phenotypes.

We also examined the effects of altering the stimulus frequency, a parameter known to affect social behaviors in males (*Hoopfer et al., 2015*). The application of a 5 Hz 0.1 mW/mm$^2$ stimulus with a 10 ms fixed duration induced significant behavioral changes over the 30 s stimulus period (*Figure 1—figure supplement 9A–C*). Stimulation at 10 Hz resulted in more extensive aggressive behavior, but further increases to 20, 30, or 50 Hz did not have a large effect. However, higher frequency stimulation did increase the amount of aggression observed during the post-stimulus period (*Figure 1—figure supplement 9D*). These experiments demonstrate that aIPg neurons promote female aggressive interactions under a range of stimulus conditions.

## aIPg neurons mediate wild-type female aggressive social interactions

The infrequent occurrence of female aggressive events under laboratory conditions (*Bath et al., 2017*; *Shelly, 1999*; *Ueda and Kidokoro, 2002*) has made it difficult to study its neuronal correlates. To facilitate such experiments, we optimized the environmental conditions. Alterations to diet and life history are known to increase female aggression in wild-type flies (*Bath et al., 2017*; *Ueda and*

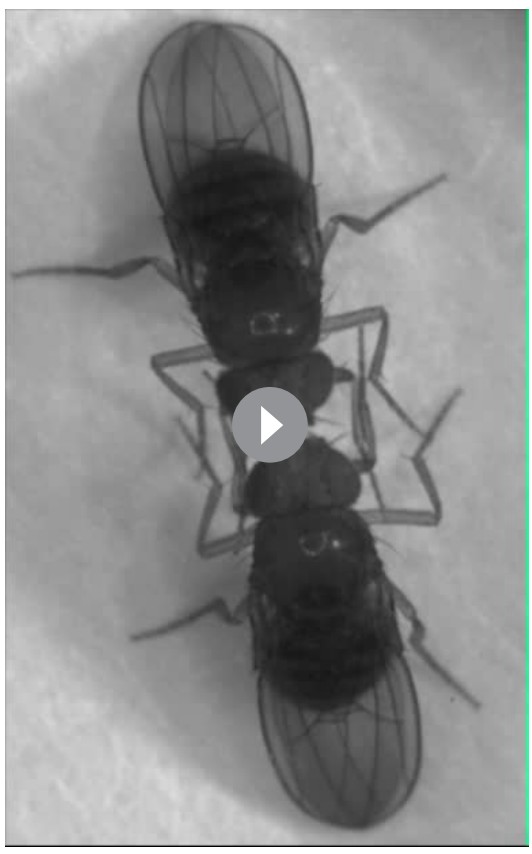

**Video 3.** High-speed video of two aIPgSS1 > Chrimson female flies.
https://elifesciences.org/articles/58942#video3

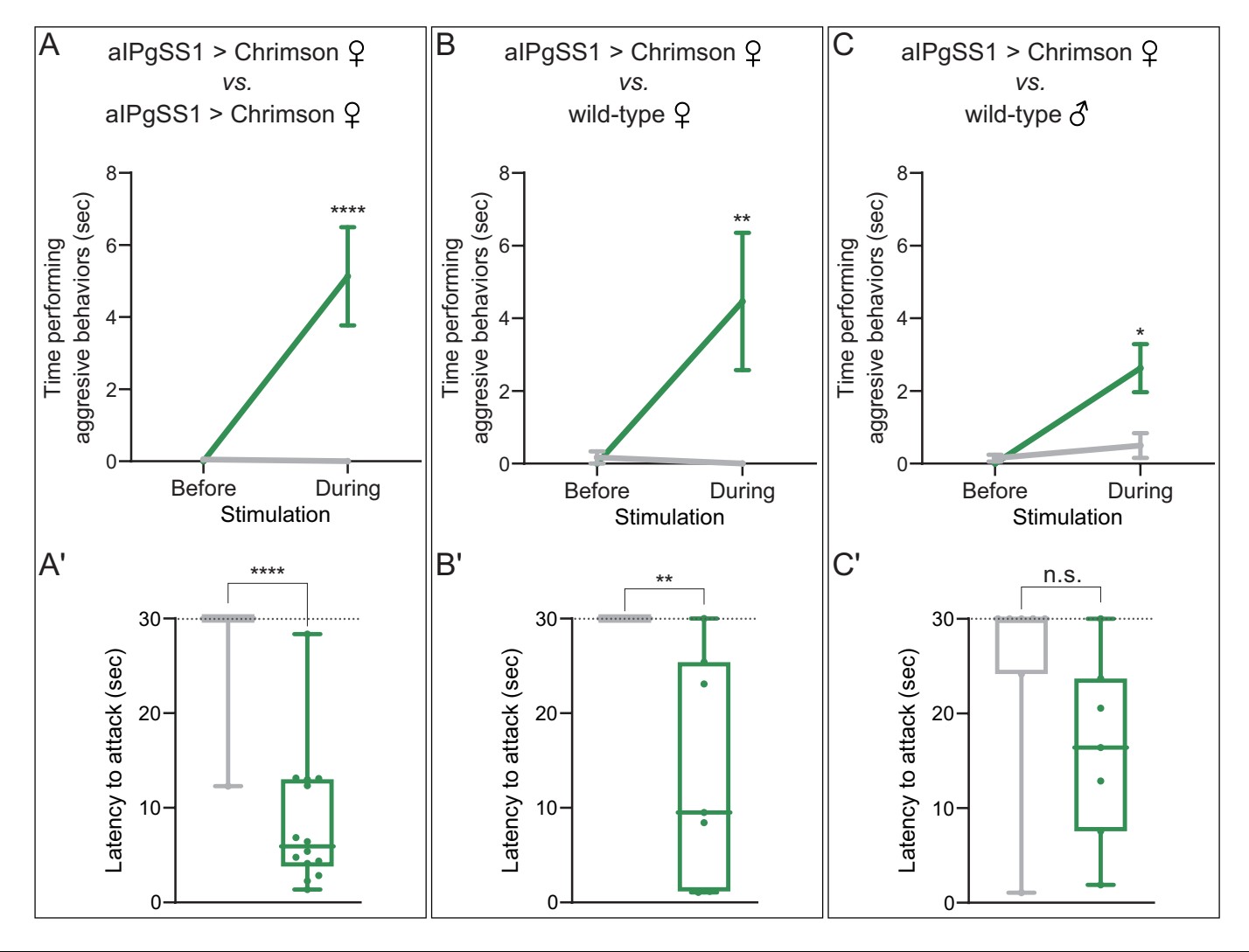

**Figure 2.** aIPg activation increases aggression against wild-type females and males. (A–C) Total time spent performing aggressive behaviors in a 16 mm arena over the 30 s period prior to or during a 0.1 mW/mm² stimulation. The plots refer only to the behavior of aIPgSS1 > Chrimson females and each arena contained only two flies: (A) two aIPgSS1 > Chrimson females; (B) an aIPgSS1 > Chrimson female and a wild-type (Canton-S) female; and (C) an aIPgSS1 > Chrimson female and a wild-type (Canton-S) male. The green line shows the stated genotype; the gray line shows the results when EmptySS > Chrimson was used instead of aIPgSS1 > Chrimson. Note that the difference observed between aggression against wild-type females (B) and males (C) was not significant (p=0.65). (A'–C') Amount of time during a 30 s 0.1 mW/mm² continuous stimulation period until first aggressive encounter. Points indicate individual flies. Dotted lines indicate the end of the trial and error bars in A–C are mean ± S.E.M. Box-and-whisker plots show median and IQR; whiskers show range. Data supporting the plots shown in the individual panels were as follows: (A') EmptySS > 20xUAS-Chrimson, n = 22 flies; aIPgSS1 > 20xUAS-Chrimson, n = 14 flies. (B') EmptySS > 20xUAS-Chrimson, n = 8 flies; aIPgSS1 > 20xUAS-Chrimson, n = 7 flies. (C') EmptySS > 20xUAS-Chrimson, n = 7 flies; aIPgSS1 > 20xUAS-Chrimson, n = 7 flies. We performed at least two biological repeats that confirmed aggression against all three types of target flies; results for a typical repeat had the following p-values during stimulation: A: p=0.0035; A': p=0.0067; B: p=0.0006; B': p=0.0157; C: p=0.0123; C': p=0.0030. A Mann-Whitney U post hoc test was used for statistical analysis. Asterisk indicates significance from 0: *p<0.05; **p<0.01; ****p<0.0001; n.s., not significant.

The online version of this article includes the following source data and figure supplement(s) for figure 2:

**Source data 1.** Source data for behavioural experiments in *Figure 2*.
**Figure supplement 1.** Increased aggression against wild-type females and males reproduced with a second aIPgSS line.
**Figure supplement 2.** aIPg activation in the absence of a target fly does not alter velocity.
**Figure supplement 3.** aIPg activation does not significantly alter copulation latency.

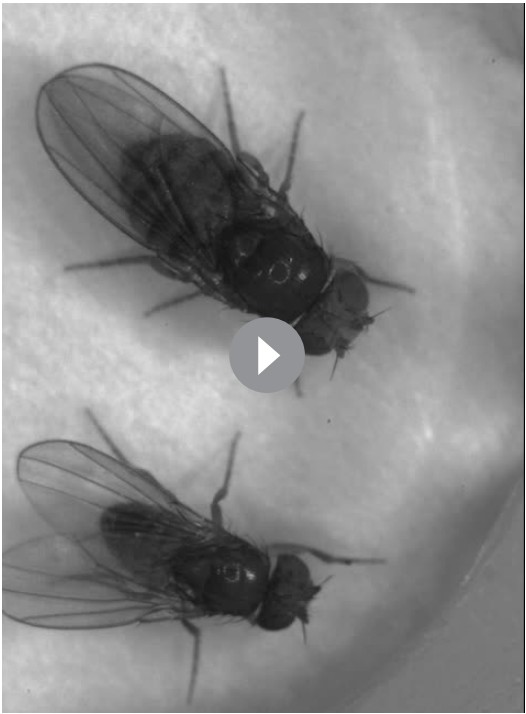

**Video 4.** High-speed video of an aIPgSS1 > Chrimson female fly with a wild-type male.
https://elifesciences.org/articles/58942#video4

*Kidokoro, 2002*). We therefore adjusted the diet of the flies to restrict protein for 20–24 hr prior to testing, included a 1 mm spot of yeast within the arena, and limited the arena size to 16 mm to observe interactions between pairs of flies (*Figure 1H*, *Figure 1—figure supplement 10E*). Under these conditions, we observed sufficient levels of aggression to examine the effects of inactivation of aIPg neurons with the synaptic inhibitor tetanus toxin (*Sweeney et al., 1995*). A significant reduction in the time spent performing aggressive behaviors, as measured using both manual and automated behavioral analyses, was observed with three different split-GAL4 lines (*Figure 1H*, *Figure 1—figure supplement 10* and *Figure 1—figure supplement 11A,B*). Such changes did not appear to be due to decreased movement as flies exhibited similar or higher velocity compared to controls over the 30 min trial (*Figure 1—figure supplement 11C,D*). These results indicate that aIPg neurons are important for modulating aggressive behaviors in females.

## Activation of aIPg overrides the requirement for specific environmental conditions for female aggressive behaviors

In addition to food availability, the genotype and sex of the target fly influence aggression (*Bath et al., 2020*; *Bath et al., 2018*; *Lim et al., 2014*; *Ueda and Kidokoro, 2002*; *Wohl et al., 2020*). Our previous neuronal activation experiments demonstrated aggression even in the absence of competition for food (*Figure 1F,G*). We next investigated the effects of activation status and the sex of the opponent in experiments using pairs of flies without prior food restriction or food present in the arena. Activation of aIPg neurons increased the total time spent displaying aggression and decreased the time from stimulus onset to the first aggressive event (attack latency), irrespective of whether aIPg neurons were stimulated in both of the females in the arena (*Figure 2A,B* and *Video 3*). These results were also reproduced using a second aIPg split-GAL4 line (*Figure 2—figure supplement 1A,B*). Females in which aIPg neurons were activated also displayed aggression when paired with wild-type male targets (*Figure 2C*, *Figure 2—figure supplement 1C*, and *Video 4*). In contrast, there were no obvious changes in behavior (*Video 5*) or velocity (*Figure 2—figure supplement 2*) upon aIPg activation in the absence of a target fly. Mating and aggressive behaviors share overlapping neuronal circuitry (*Asahina et al., 2014*; *Hoopfer et al., 2015*; *Watanabe et al., 2017*). We therefore examined the copulation latency following stimulation of aIPg neurons in virgin females. Copulation latency did not significantly differ from that of controls when paired with single housed, naïve, wild-type males (*Figure 2—figure supplement 3*), although we cannot rule out more subtle changes in the progression of mating behavior. Taken together, our results indicate that activation of aIPg neurons can increase the likelihood of aggression directed at both females and males irrespective

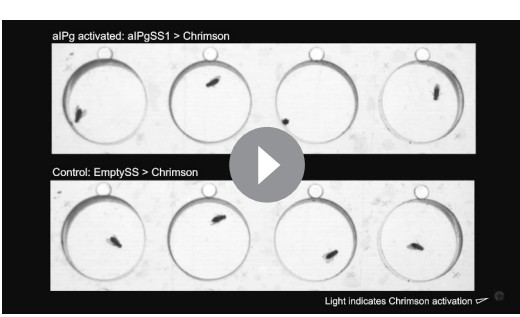

**Video 5.** Behavior of individual aIPgSS1 > Chrimson and EmptySS > Chrimson flies.
https://elifesciences.org/articles/58942#video5

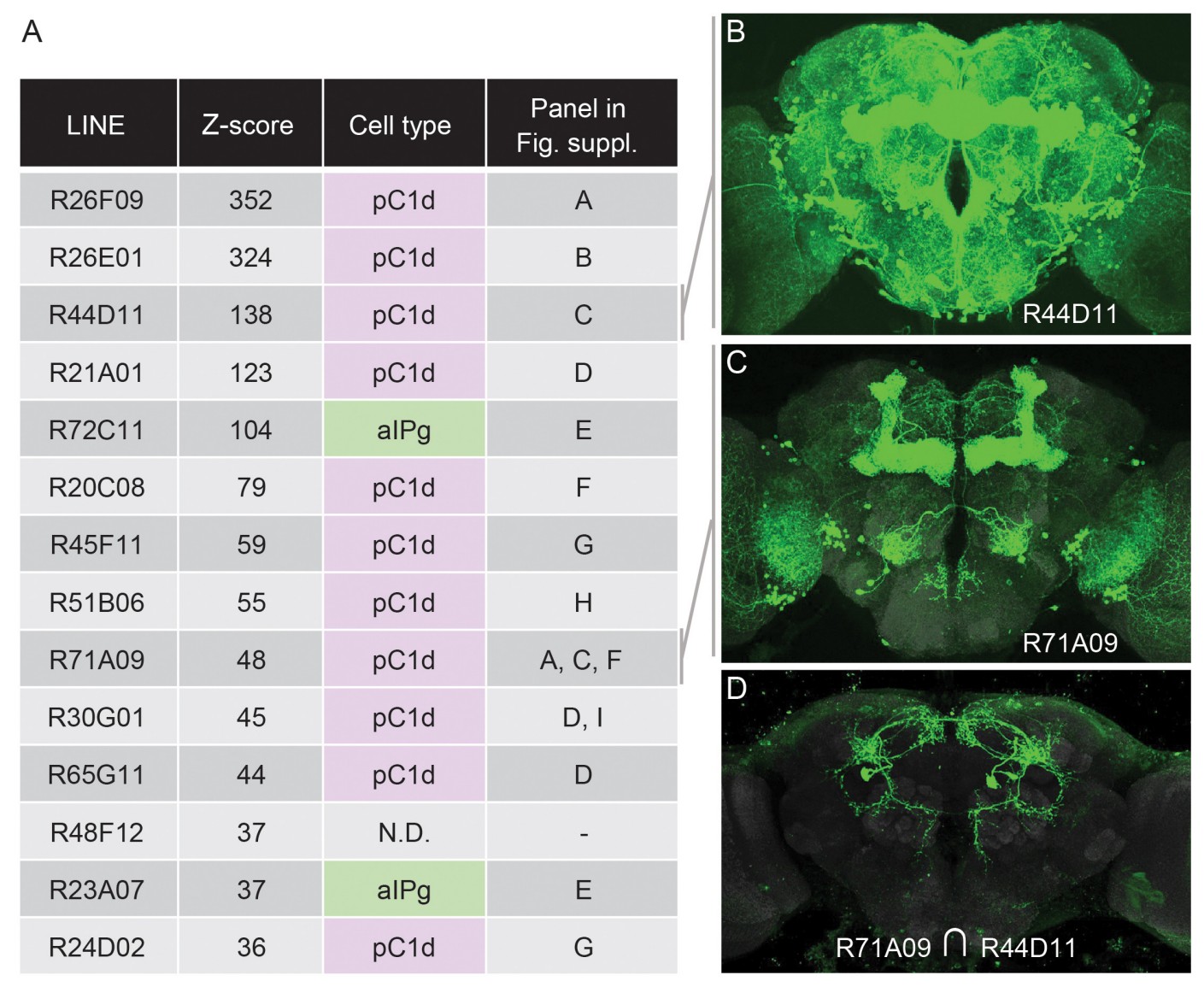

**Figure 3.** An unbiased screen suggests that aIPg1-3 and pC1d are key nodes for gating female-female aggression. (**A**) The fourteen top hits for female-female chasing from an unbiased activation screen of 2204 generation 1 GAL4 lines are listed along with their Z-scores, the signed number of standard deviations from the mean behavior of the control, as determined by *Robie et al., 2017*. Also shown is the relevant cell type we concluded from our intersectional analysis (*Figure 4—figure supplement 1*) to be present in each line. N.D., no cell type reproducibly detected, suggesting that R48F12 may not share a common cell type with any of the other 13 lines. The final column in the table refers to the panels in *Figure 3—figure supplement 1* where results supporting the stated conclusion are shown. (**B,C**) The expression patterns of the two indicated GAL4 lines (*Jenett et al., 2012*). The images shown were taken from the database at http://www.janelia.org/gal4-gen1, where the expression patterns of the other lines listed in A can also be found. (**D**) The expression pattern of a split-GAL4 line made by intersecting these two enhancers; a cell with the morphology of pC1d can be seen. The online version of this article includes the following figure supplement(s) for figure 3:

**Figure supplement 1.** Expression patterns resulting from split-GAL4 intersections of hits from the unbiased screen.

of environmental conditions known to promote aggression in wild-type females.

## Identifying additional cell types involved in mediating female aggression

Having established a role for aIPg neurons in female aggression, we used two complementary methods to discover additional cells involved in regulating this behavior. First, we used behavioral screens

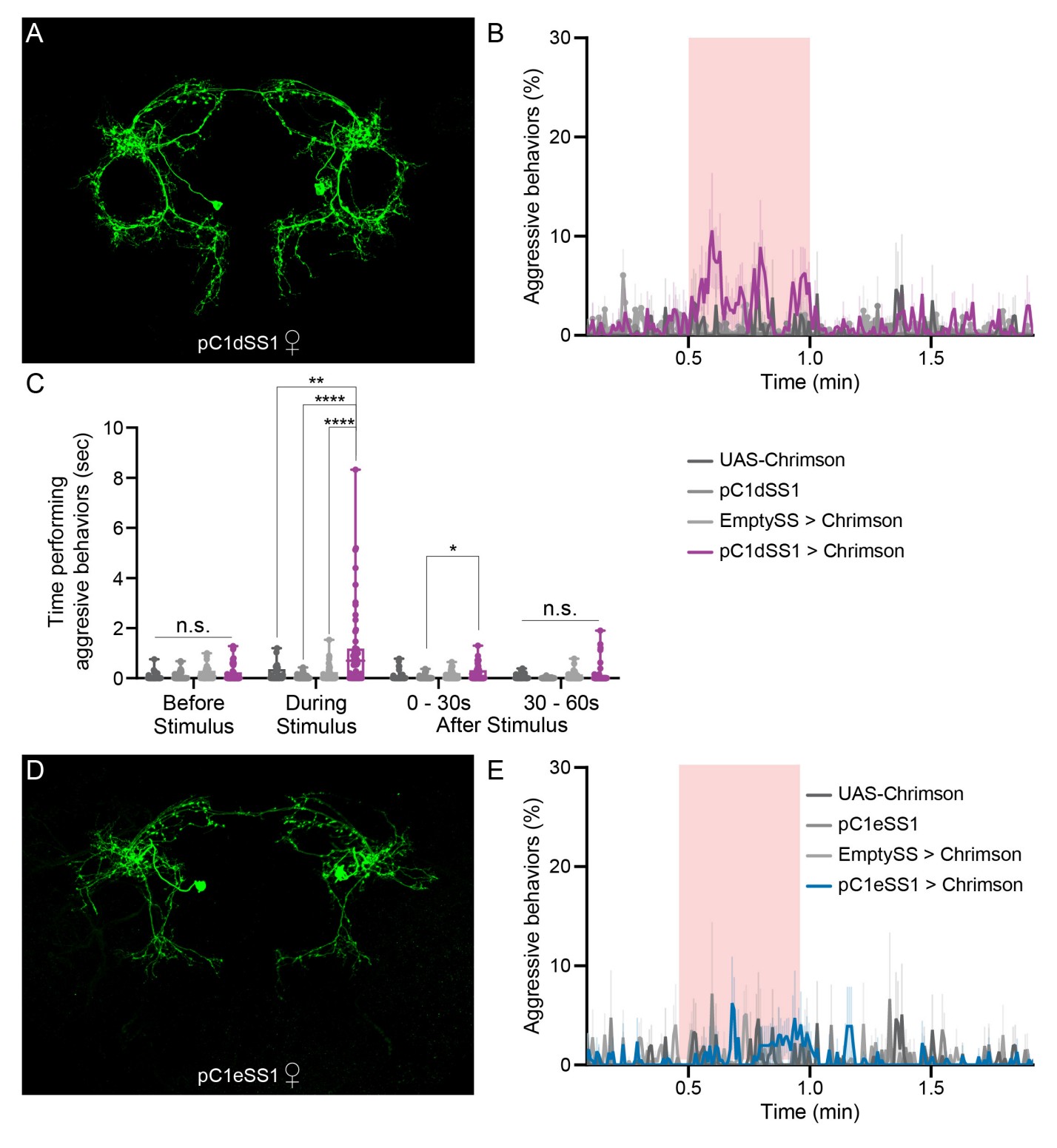

**Figure 4.** pC1d, but not pC1e, significantly increases aggressive social interactions in female flies. (**A**) MIP (63x) image of the central brain of a female from the pC1dSS1 split-GAL4 line crossed with 20xUAS-CsChrimson::mVenus and stained with anti-GFP antibody. Images of the complete brain and ventral nerve cord of a female and male of the same genotype are shown in *Figure 4—figure supplement 1A–C*. (**B**) Percentage of flies engaging in aggressive behaviors over the course of a trial during which a 30 s 0.4 mW/mm² continuous light stimulus (pink shading) was delivered, plotted as in *Figure 1F*. (**C**) Total time an individual spent performing aggressive behaviors during each of four 30 s periods: prior to, during, immediately following,

*Figure 4 continued on next page*

*Figure 4 continued*

and 30–60 s after the stimulus. Points represent individual flies. Note that we used 20xUAS-CsChrimson for these experiments to be consistent with the experiments done with aIPg split-GAL4 lines, but the levels of aggression observed with pC1dSS1 are actually higher when a weaker effector line (5xUAS-CsChrimson) is used (*Figure 4—figure supplement 9A*). Box-and-whisker plots show median and IQR; whiskers show range. Kruskal-Wallis and Dunn's post hoc tests were used for statistical analysis. Asterisk indicates significance from 0: *p<0.05; **p<0.01; ****p<0.0001; n.s., not significant. (D) MIP (63x) image of the central brain of a female from the pC1eSS1 line crossed with 20xUAS-CsChrimson::mVenus and stained with anti-GFP antibody. Images of the complete brain and ventral nerve cord of a female and male of the same genotype are shown in *Figure 4—figure supplement 2A–C*. (E) Percentage of flies engaging in aggressive behaviors over the course of a trial during which a 30 s 0.4 mW/mm$^2$ continuous light stimulus (pink shading) was delivered, plotted as in *Figure 1F*. No obvious differences between genotypes were observed. Data supporting the plots shown in the individual panels were as follows: B: 20xUAS-CsChrimson, n = 2 experiments; pC1dSS1, n = 3 experiments; EmptySS > 20xUAS-CsChrimson, n = 5 experiments; pC1dSS1 > 20xUAS-CsChrimson, n = 5 experiments. C: 20xUAS-CsChrimson, n = 29 flies; pC1dSS1, n = 48 flies; EmptySS > 20xUAS-CsChrimson, n = 46 flies; pC1dSS1 > 20xUAS-CsChrimson, n = 53 flies. E: 20xUAS-CsChrimson, n = 2 experiments; pC1eSS1, n = 2 experiments; EmptySS > 20xUAS-CsChrimson, n = 3 experiments; pC1eSS1 > 20xUAS-CsChrimson, n = 3 experiments. For all panels, data are representative of at least three independent biological repeats, one of which is shown here; see *Supplementary file 3* for exact p-values.

The online version of this article includes the following source data and figure supplement(s) for figure 4:

**Source data 1.** Source data for behavioural experiments in *Figure 4*.
**Figure supplement 1.** Expression patterns of pC1d split-GAL4 lines.
**Figure supplement 2.** Expression patterns of pC1e split-GAL4 lines.
**Figure supplement 3.** Morphologies of individual pC1d neurons.
**Figure supplement 4.** Morphologies of individual pC1e neurons.
**Figure supplement 5.** Behavioral characterization of female flies after pC1d activation.
**Figure supplement 6.** Optogenetic activation of additional lines labeling pC1d split-GAL4 lines display similar behavioral results to pC1dSS1.
**Figure supplement 7.** Optogenetic stimulation of pC1dSS1 > Chrimson males does not result in aggressive behavior.
**Figure supplement 8.** Optogenetic activation of aggression depends on feeding all *trans*-retinal.
**Figure supplement 9.** Behavioral effects of stimulus delivery and effector strength.
**Figure supplement 10.** Behavioral effects of the frequency of optogenetic stimulation.
**Figure supplement 11.** pC1d activation also increases aggression against wild-type females and males.
**Figure supplement 12.** pC1d inactivation did not significantly diminish aggressive behavior.
**Figure supplement 13.** Optogenetic activation of additional lines labeling pC1e.
**Figure supplement 14.** Behavioral effects of stimulus delivery and effector copy number.
**Figure supplement 15.** Comparison of activation phenotypes of pC1d, pC1e and pC1a-c.

to identify other cell types that could drive female aggression when activated. Second, we used the aIPg neurons as an entry point for EM-based circuit mapping. As described below, both approaches converged on the same set of cells.

## aIPg and pC1d are two key groups of neurons involved in female aggressive behaviors

Other neurons in the same circuit as the aIPg neurons, or in parallel pathways, might also be able to induce aggression when activated. To identify such neurons, we took a strategy analogous to that used by geneticists to ask how many different genes can produce a particular phenotype when mutated. In that strategy, individual mutations are placed into complementation groups after performing a genetic screen large enough to sample all genes. In this way, the number of different genes that can give rise to the phenotype under study when mutated can be estimated (see for example, *Nüsslein-Volhard and Wieschaus, 1980*). To carry out an analogous approach, we started with lines identified as having increased female-female chasing behavior in a previous screen of over 2000 GAL4 lines (*Robie et al., 2017*). Each of the GAL4 lines used in that screen (*Jenett et al., 2012*) had broad expression, precluding the identification of the specific cell types

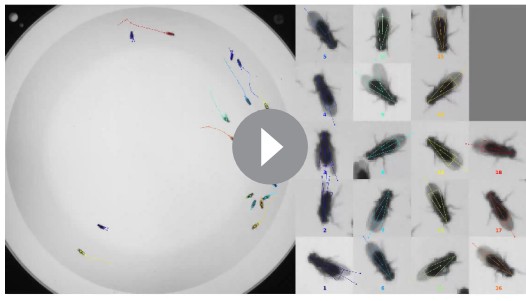

**Video 6.** Behavior of pC1dSS1 > Chrimson flies.
https://elifesciences.org/articles/58942#video6

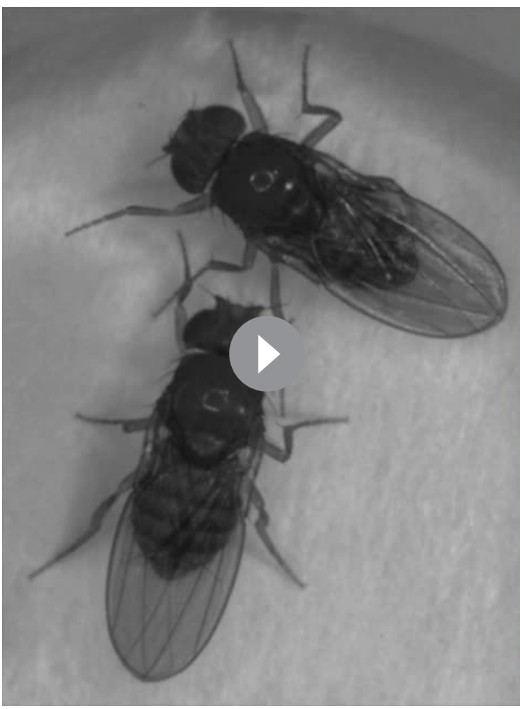

**Video 7.** High-speed video of two pC1dSS1 > Chrimson female flies.
https://elifesciences.org/articles/58942#video7

responsible for the observed behavior. To identify these cell types, we generated split-GAL4 hemidriver lines using the enhancers from this screen's top hits and then crossed them to each other to reveal the presence of shared cell types (*Figure 3*; *Figure 3—figure supplement 1*). Strikingly, 13 of the top 14 hits identified by their behavioral score could be accounted for by just two cell types: aIPg neurons and pC1d, one of the five cell types in the female pC1 cell cluster. While such screens could miss cell types with less penetrant activation phenotypes and those that inhibit aggression, these results imply central roles for aIPg and pC1d in female aggression.

The five cell types that make up the pC1 group express *dsx* and have been implicated in female receptivity, oviposition, male courtship, and both male and female aggression (*Deutsch et al., 2020*; *Fathy, 2016*; *Hoopfer et al., 2015*; *Ishii et al., 2020*; *Koganezawa et al., 2016*; *Palavicino-Maggio et al., 2019*; *Rideout et al., 2010*; *Wang et al., 2020a*; *Wang et al., 2020b*; *Wohl et al., 2020*; *Zhou et al., 2014*). In previous work, the lack of cell type specific genetic reagents has made it difficult to elucidate the relative contribution of the five pC1 cell types in females to each of these behaviors. pC1d cells are identified in intersections using the enhancers from 11 of the top 14 hits from the *Robie et al., 2017* screen (*Figure 3*; *Figure 3—figure supplement 1*). However, nearly all of these intersections contain at least one other pC1 cell type, leaving open the possibility that inducing aggression requires a combination of multiple pC1 cell types.

## Activation of pC1d alone, but not pC1e or pC1a - c, promotes female aggressive behaviors

To address the role of individual pC1 cell types in female aggression, we generated split-GAL4 lines that drive expression in either only pC1d or pC1e, as well as lines containing both cell types (*Figure 4A,D*; *Figure 4—figure supplements 1*, *2*, *3* and *4*). We also used a split-GAL4 line that labels pC1a - c (provided by K. Wang and B. Dickson). No expression was observed in males in the majority of the lines used (*Figure 4—figure supplement 1C* and *Figure 4—figure supplement 2C*).

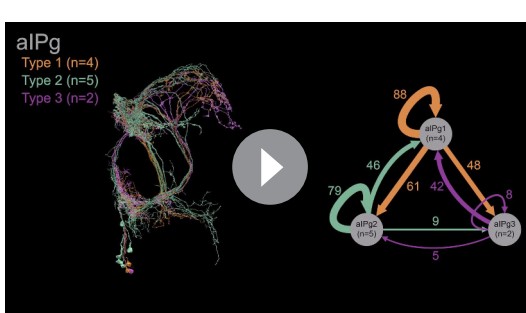

**Video 8.** Interconnectivity of aIPg types 1–3 neurons.
https://elifesciences.org/articles/58942#video8

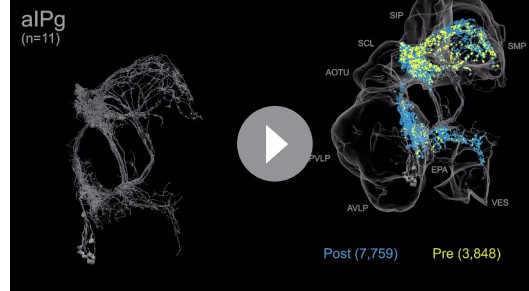

**Video 9.** Locations of pre- and post-synaptic connections with aIPg type 1–3 neurons.
https://elifesciences.org/articles/58942#video9

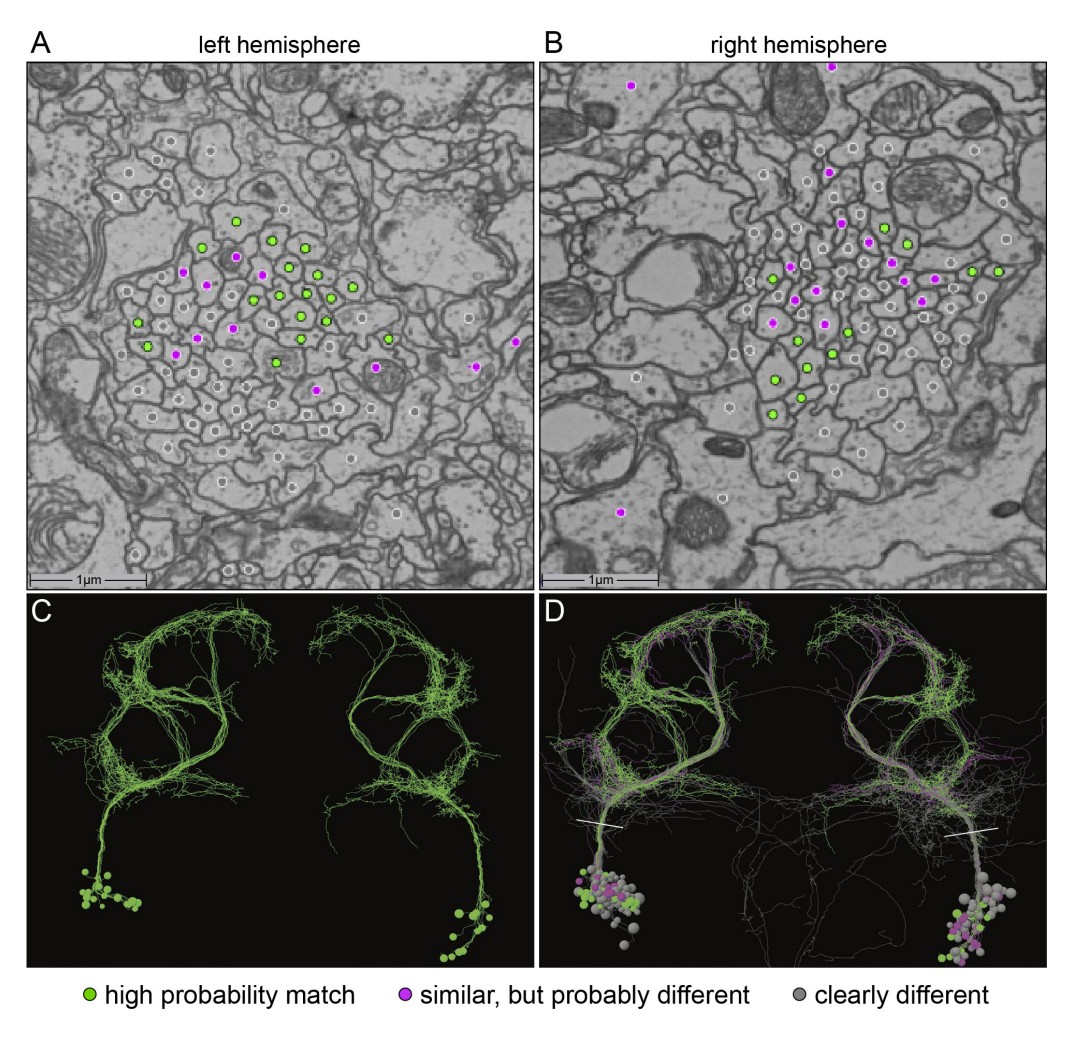

**Figure 5.** Identification of aIPg neurons in the FAFB dataset. (**A,B**) Images of an area of the left and right hemisphere of an EM section from the FAFB dataset containing the fiber tracts of the putative aIP-g neurons described by *Cachero et al., 2010*. A dot has been placed in each axon, color-coded to reflect the degree of similarity of its morphology, revealed by manual tracing and visual inspection, to the aIPg neurons contained in our split-GAL4 lines. Gray represents neurons whose morphology clearly differed, magenta represents neurons whose morphology were similar but differed in one or more branches, and green represents neurons that we judged to correspond to those in our split-GAL4 lines. Note that, as we often observe in our split-GAL4 lines (see *Figure 1—figure supplement 1*) and has been reported for many cell types in connectomic studies (see for example, *Bates et al., 2020*), the number of neurons often differs between hemispheres. In this brain, the left hemisphere had 17 green cells, while the right hemisphere had only 12. (**C,D**) Skeleton rendering of the traced aIPg neurons, colored based on their similarity to the aIPg neurons identified in our split-GAL4 lines. Panel C shows only cells we judged to correspond to those in our split-GAL4 lines, while D shows all traced cells. White lines in D indicate the approximate plane of the images shown in A and B. Tracing of gray neurons was stopped when it was clear they did not match our split-GAL4 lines; therefore, their arbors are likely to be incomplete in these images.

The online version of this article includes the following figure supplement(s) for figure 5:

**Figure supplement 1.** Identification of aIPg neurons in the hemibrain dataset.

The cells labeled in males in two of the lines were not morphologically similar to pC1d or pC1e and these lines were not used for behavioral analysis in males. RNAseq analysis of pC1d and pC1e neurons confirmed that the cells expressed *dsx* and the use of neurotransmitter acetylcholine (*Figure 1—figure supplement 5*), as previously described (*Palavicino-Maggio et al., 2019*; *Rezával et al., 2016*; *Rideout et al., 2010*; *Zhou et al., 2014*). Our data suggest that these cells might also express *fru* (*Figure 1—figure supplement 5*).

Using the same stimulation parameters as we used for lines labeling aIPg neurons, optogenetic activation of pC1d alone increased the percentage of aggressive flies (*Figure 4B,C*, *Figure 4—*

*figure supplement 5A* and *Video 6*). These aggressive interactions included fencing and head butting (*Video 7*) and were accompanied by a slight decrease in walking (*Figure 4—figure supplement 5B*) as well as increased touching and chasing (*Figure 4—figure supplement 5C,D*). Similar results were seen with additional split-GAL4 lines labelling pC1d (*Figure 4—figure supplement 6*). Behavioral changes were not observed upon stimulation of males (*Figure 4—figure supplement 7*). Additional controls in which all *trans*-retinal was omitted from the food did not show an elevation in aggression following light onset (*Figure 4—figure supplement 8*). Increasing the intensity of stimulation under conditions of constant illumination did not significantly heighten behavior (*Figure 4—figure supplement 9B*) suggesting that the lowest intensity (0.04 mW) of stimulating light used was already saturating. Moreover, changes in the expression level of CsChrimson were inversely correlated with behavior (*Figure 4—figure supplement 9A*), as was increasing stimulus intensity from 0.1 to 0.4 mW in flies expressing 20xUAS-CsChrimson (*Figure 4—figure supplement 9C*), suggesting that higher effector expression, or activation levels, might be detrimental to cell function. However, increasing the number of individual light pulses, while shortening their length so as to maintain a constant total stimulus duration, resulted in more aggression during the stimulus period. We also found a slight increase in aggression following stimulation at 50 Hz, the highest pulse frequency tested (*Figure 4—figure supplement 10*).

As we observed with aIPg, pC1d stimulation also resulted in a decrease in the attack latency and increase in the time spent performing aggressive behaviors irrespective of whether the opponent was an activated female, wild-type female or male (*Figure 4—figure supplement 11*). While activation promoted aggression, no differences were observed following pC1d inactivation with tetanus toxin (*Figure 4—figure supplement 12*). These results suggest that pC1d neurons are not essential for female aggression; however, we have not independently confirmed the degree of effectiveness of the tetanus toxin inactivation.

In contrast to pC1d, stimulation of lines containing pC1e alone did not significantly alter any of the behaviors we assayed under a variety of conditions (*Figure 4D,E*; *Figure 4—figure supplement 13*; *Figure 4—figure supplement 14*). Analysis of lines containing both pC1d and pC1e exhibited similar levels of behavior to lines containing pC1d alone, implying that pC1d and pC1e do not act synergistically (*Figure 4—figure supplement 6*). Likewise, activation of pC1a - c did not change the percentage of flies displaying aggression (*Figure 4—figure supplement 15*). Taken together, our results suggest that pC1d, but not pC1e or pC1a - c, acts as a significant facilitator of female aggression.

## aIPg and pC1d neurons are interconnected, but have largely distinct upstream and downstream partners

The generation of the full adult female brain (FAFB) electron microscopic (EM) image set (*Zheng et al., 2018*) and the connectome of the hemibrain (*Scheffer et al., 2020*) allowed us to use EM-level connectomics to determine the structure of the circuit(s) that contained the cells identified through our behavioral studies. First, we identified the cells in EM volumes that correspond to those observed in our aIPg split-GAL4 lines. We began this work in FAFB, before the availability of the

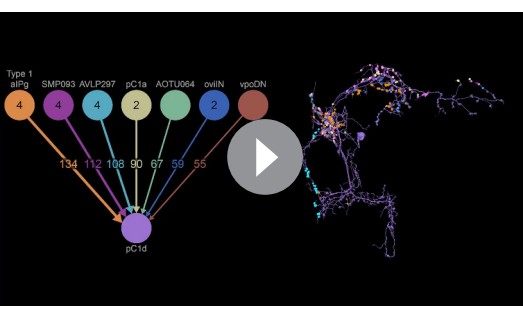

**Video 10.** Top inputs to pC1d.
https://elifesciences.org/articles/58942#video10

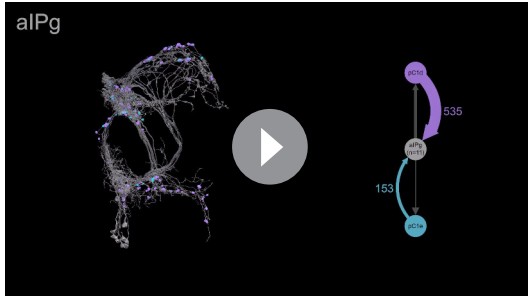

**Video 11.** Interconnectivity of aIPg types 1–3, pC1d and pC1e neurons.
https://elifesciences.org/articles/58942#video11

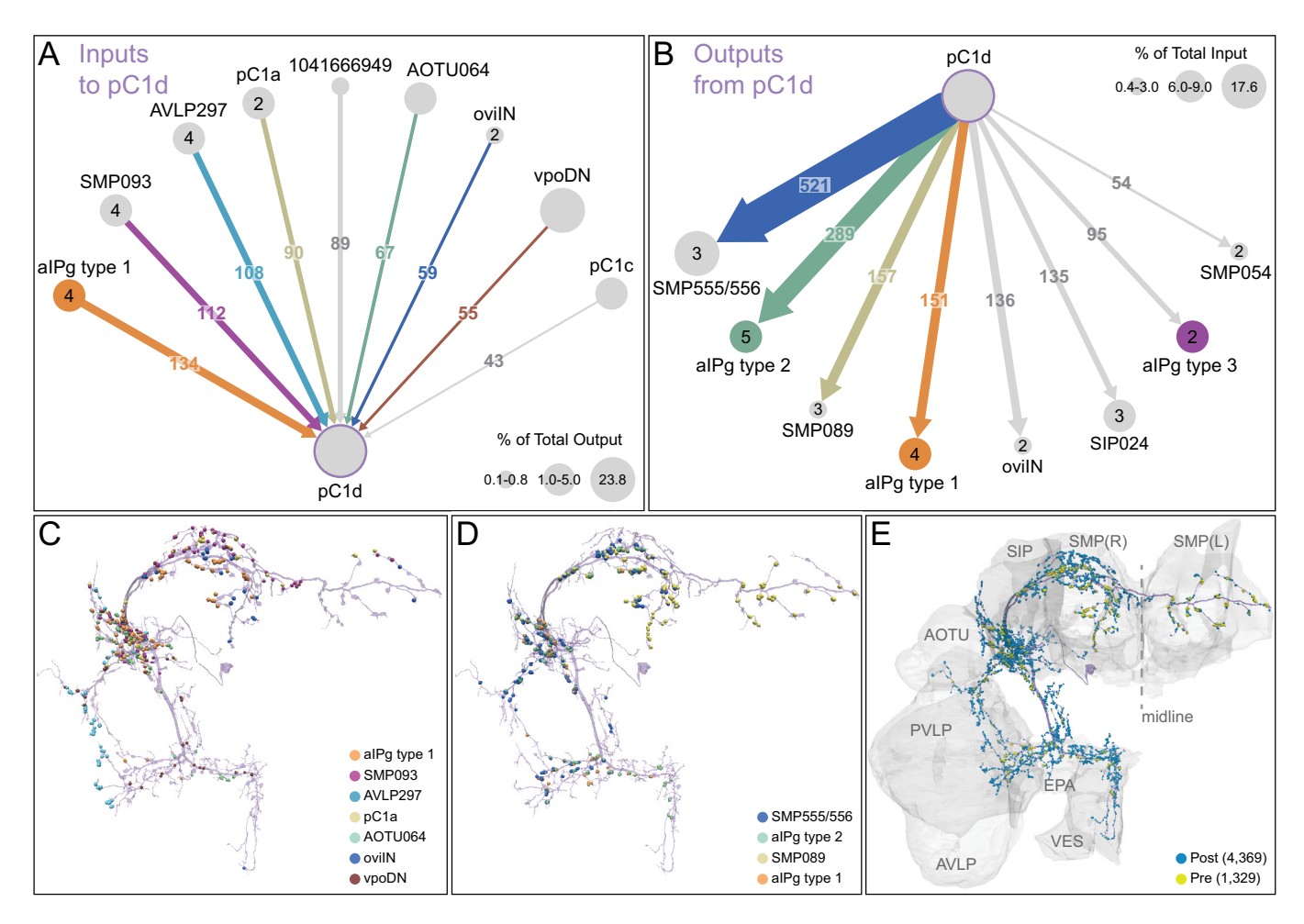

**Figure 6.** Major inputs to and outputs from pC1d. (**A**) Pre-synaptic inputs to the right-hemisphere pC1d neuron. All neurons making more than 100 synapses are shown, along with other neurons selected based on biological interest. Numbers in the arrows represent synapse number. The size of the circles representing input neurons indicates the estimated percentage of their output (number of synapses to pC1d/total output synapse number in the hemibrain volume to all 'traced' neurons) that goes to pC1d; note that vpoDN is a descending interneuron with outputs in the ventral nerve cord, which lies outside the hemibrain volume. Numbers in the circles indicate the number of neurons of that cell type present in the hemibrain, if greater than one. Note that, while there is only one pC1a neuron per hemisphere, both the left and right-hemisphere pC1a neurons make synapses onto the right-hemisphere pC1d. (**B**) Post-synaptic outputs of the single right-hemisphere pC1d. All neurons receiving more than 100 synapses from pC1d are shown, along with other neurons selected based on biological interest. Numbers in the arrows represent synapse number. The size of the circles representing output neurons indicates the percentage of their input (estimated by synapse number) that comes from pC1d. (**C**) Positions on the pC1d arbor of post-synaptic sites, color-coded to match diagram in (**A**). (**D**) Positions on the pC1d arbors of the presynaptic sites where the connections diagrammed in (**B**) occur, color-coded. (**E**) Positions of pC1d's pre- (yellow) and post-synaptic (blue) connections to all neurons are shown; relevant brain areas are indicated. See *Video 10* for better visualization of the inputs to pC1d and *Video 12* for better visualization of pC1d's outputs.

hemibrain dataset. The fiber bundle in each hemisphere that contained the neurons corresponding to the aIP-g lineage was first identified based on their cell body and soma tract position in the brain (*Cachero et al., 2010*). We then traced the major arbors of all these cells sufficiently to determine if they matched the morphologies of the neurons observed in our split-GAL4 lines. Twelve of the putative aIP-g neurons found in the right hemisphere resembled those in our split-GAL4 lines (*Figure 5*). During the process of generating the hemibrain connectome, we were also able to identify a set of eleven aIPg cells corresponding to those found in FAFB (*Figure 5—figure supplement 1*) as well as neurons in the pC1 cluster. The emergence of hemibrain dataset allowed us to both analyze their morphology in greater detail and participate in the effort to improve the accuracy of their reconstruction (*Scheffer et al., 2020*). Based on morphological differences in their projections, later

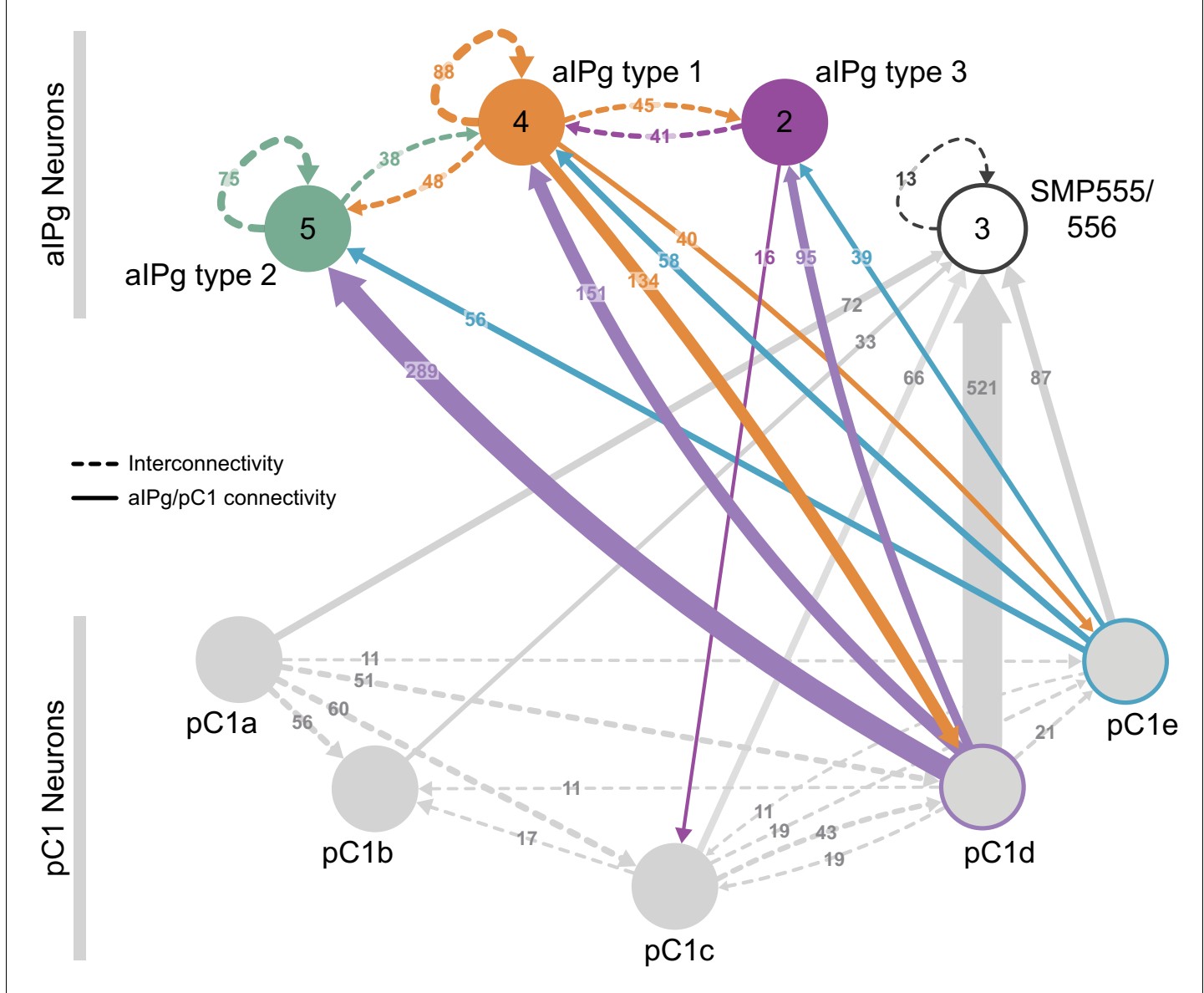

**Figure 7.** Reciprocal connections between aIPg and pC1 neurons. (**A**) Interconnectivity between the right-hemisphere pC1a-e, aIPg type 1–3, and SMP555/556 neurons; only connections with 10 or more synapses are shown. SMP555/556 neurons also appear to derive from the aIP-g lineage (see *Figure 5—figure supplement 1*). Synapse number is noted on each arrow and dashed lines represent interconnectivity within the aIPg type 1, type 2 or type 3, SMP555/556 or pC1a – e neurons. Numbers in the circles indicate the number of neurons within the cell type present in the right brain hemisphere, if greater than one. Arrows are color-coded to correspond to the presynaptic cell type. See *Video 8* for additional morphological detail on interconnectivity among aIPg neuron types and *Video 11* for additional morphological detail on the connections between aIPg and pC1 neurons.

confirmed by connectivity, we further separated the aIPg neurons into three distinct—but interconnected—types, aIPg type 1, aIPg type 2, and aIPg type 3 (*Video 8*) that have pre- and postsynaptic sites intermingled throughout their arbors (*Video 9*). Despite extensive efforts, we were unable to generate genetic tools to separately manipulate each of these aIPg types. Thus, all of our behavioral results reflect activation or inactivation of the combined populations of these three related cell types. A fourth set of neurons is named aIPg type 4 in the hemibrain v1.1 database, but these neurons were not considered here due to their distinct morphology and connectivity. Likewise, the three SMP555/556 neurons, while bearing a morphological resemblance to aIPg neurons, are clearly distinct in their projections and connectivity (*Figure 5—figure supplement 1*). Neither aIPg type 4 nor

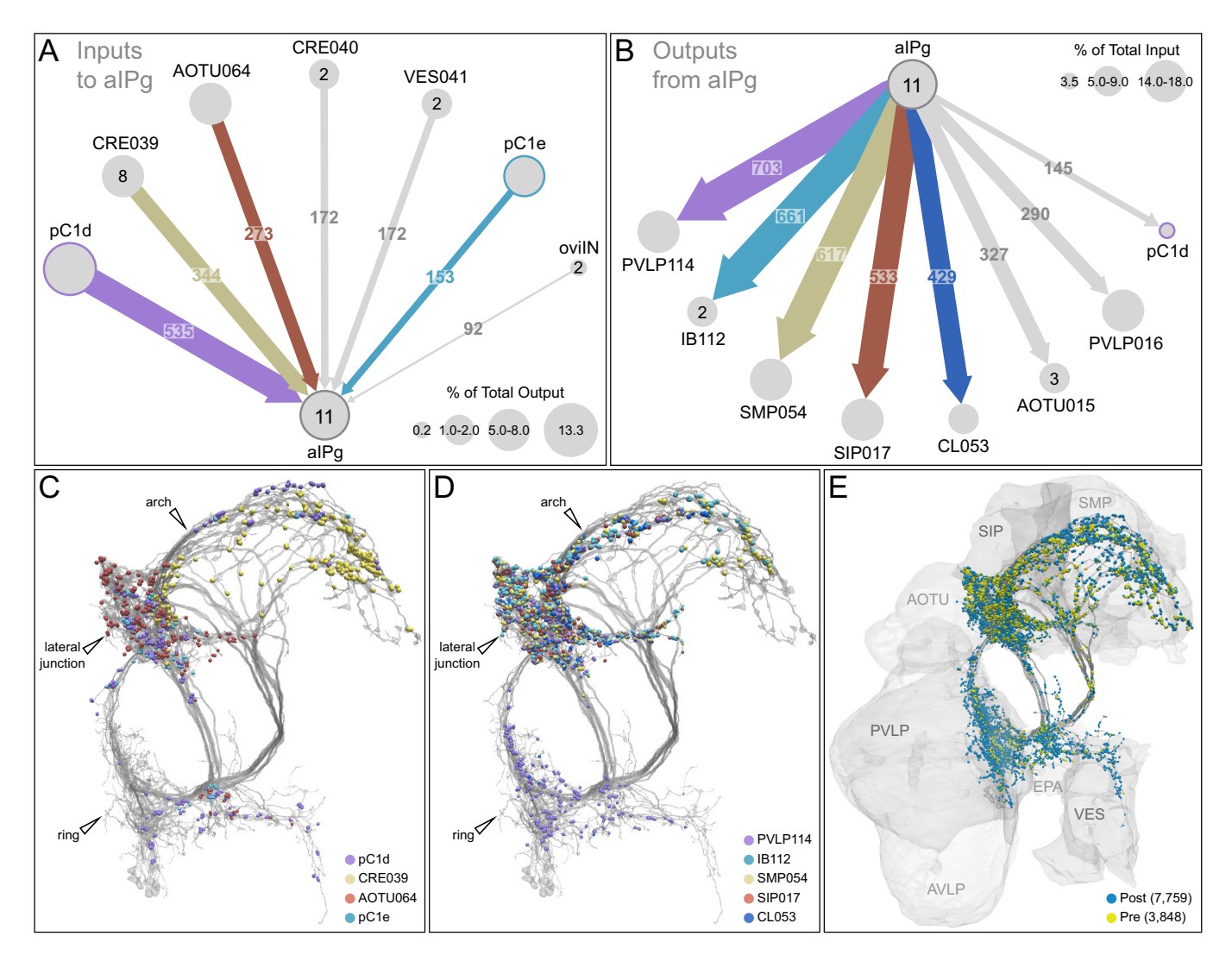

**Figure 8.** Major inputs to and outputs from aIPg neurons. (**A**) Inputs to the 11 aIPg type 1–3 neurons in the right hemisphere; only connections with 150 or more synapses are shown and inputs to all 11 aIPg neurons have been pooled. OviIN was included, even though it did not meet this threshold, due to its involvement in other female behaviors. The size of the circles representing input neurons indicates the percentage of their output (estimated by synapse number) that goes to aIPg neurons. Numbers in the circles indicate the number of neurons of that cell type present, if greater than one. The number of cells and synapses given for a cell type include neurons in the left hemisphere if they are connected to the right-hemisphere aIPgs, with the exception of pC1d where only the right-hemisphere pC1d is shown; pC1d_L makes 71 synapses to aIPgs. (**B**) Post-synaptic outputs of aIPg neurons; only connections with 290 or more synapses are shown except for pC1d (145 synapses), which has been included to point out reciprocal connections. The size of the circles representing the downstream targets of aIPg indicates the percentage of their input (estimated by synapse number) that comes from aIPg neurons. Only one of the top downstream targets of the aIPg neurons, SMP054, also receives strong input from pC1d, although the connectivity strengths differ considerably between the two, with 14.4% and 1.1% of its inputs provided by aIPg and pC1d neurons, respectively. (**C**) Positions on the aIPg arbors of post-synaptic sites where the connections diagrammed in (**A**) occur, color-coded. (**D**) Positions on the aIPg arbors of the presynaptic sites where the connections diagrammed in (**B**) occur, color-coded. (**E**) Positions of aIPg's pre- (yellow) and post-synaptic (blue) connections to all neurons are shown; relevant brain areas are indicated. See *Video 13* for better visualization of the inputs to aIPg neurons and *Video 14* for better visualization of their outputs.

SMP555/556 neurons appear to be contained in our split-GAL4 lines (*Figure 1*, *Figure 1—figure supplement 2*).

We next identified each of the five pC1 cell types in the hemibrain volume. As no other cells with similar morphology were found in the hemibrain volume, we are confident in the correspondence of pC1 cell types between the connectome and our split-GAL4 lines (*Figure 4—figure supplement 3*,

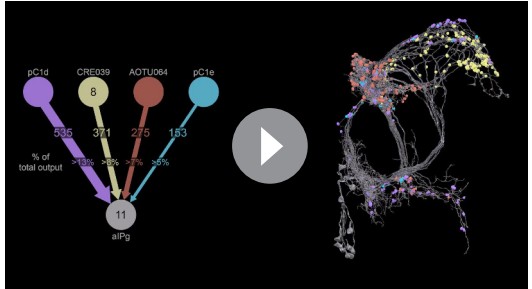

**Video 12.** Top outputs from pC1d.
https://elifesciences.org/articles/58942#video12

**Video 13.** Top inputs to aIPg types 1–3 neurons.
https://elifesciences.org/articles/58942#video13

*Figure 4—figure supplement 4*). Moreover, our light to EM-level assignments, made in the hemibrain volume, are consistent with those reported by *Wang et al., 2020a* using FAFB. Our subsequent connectivity analysis focused on pC1d and aIPg types 1–3 as they are the cell types for which we established a clear role in female aggression.

The inputs and outputs of the aIPg and pC1d neurons gave us clues about how other sensory and behavioral information might be integrated to influence the probability of engaging in aggressive behaviors. For example, inputs to pC1d provide potential paths by which the status of several other behaviors might be conveyed (*Figure 6A*, *Video 10*), including cell types implicated in mating, oviposition (oviIN, pC1a), and receptivity (vpoDN) (*Wang et al., 2020a*; *Wang et al., 2020b*). The descending interneuron vpoDN sends 24% of its synaptic output in the central brain to pC1d (*Figure 6A*, *Supplementary file 2*). The aIPg types 1, 2 and 3, pC1d and pC1e are highly interconnected (*Figure 7*, *Video 11*). Reciprocal connectivity was observed between pC1d and aIPg type 1 neurons, while pC1d delivers unidirectional input to aIPg types 2 and 3, and SMP555/556 neurons (*Figure 6A–D*, *Figure 7*, *Videos 11* and *12*). The top inputs to and outputs from the aIPg neurons are shown in *Figure 8* and in more detail in *Videos 13* and *14*.

The vast majority of the inputs and outputs of the aIPg neurons and pC1d are distinct. For, example, if we compare cell types connected by 20 or more synapses only eight of the 52 cell types that are presynaptic to pC1d are also presynaptic to any of the aIPg 1–3 cell types. Similarly, only 11 of pC1d's 37 postsynaptic targets are also postsynaptic to any of the aIPg cell types. These differences imply that, while both aIPg and pC1d are sexually dimorphic and play a role in female aggression, they also participate in a number of nonshared functions.

The execution of social behaviors requires sensory information from the surrounding environment to be processed and integrated with internal state. Changes in internal state can alter the way in which sensory information is processed and motor outputs are executed (*Chen and Hong, 2018*). The large number of downstream targets of the aIPg neurons—84 different cell types receive 20 or more synapses from at least one of the three

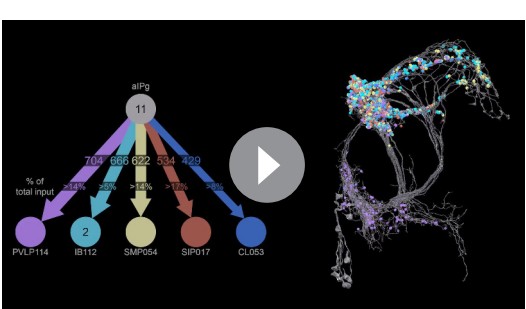

**Video 14.** Top outputs from aIPg types 1–3 neurons.
https://elifesciences.org/articles/58942#video14

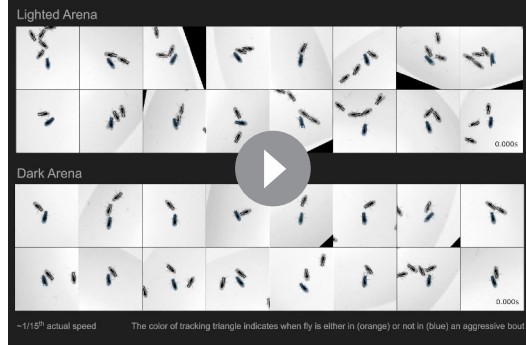

**Video 15.** Comparison of aggressive bouts under lighted and dark conditions.
https://elifesciences.org/articles/58942#video15

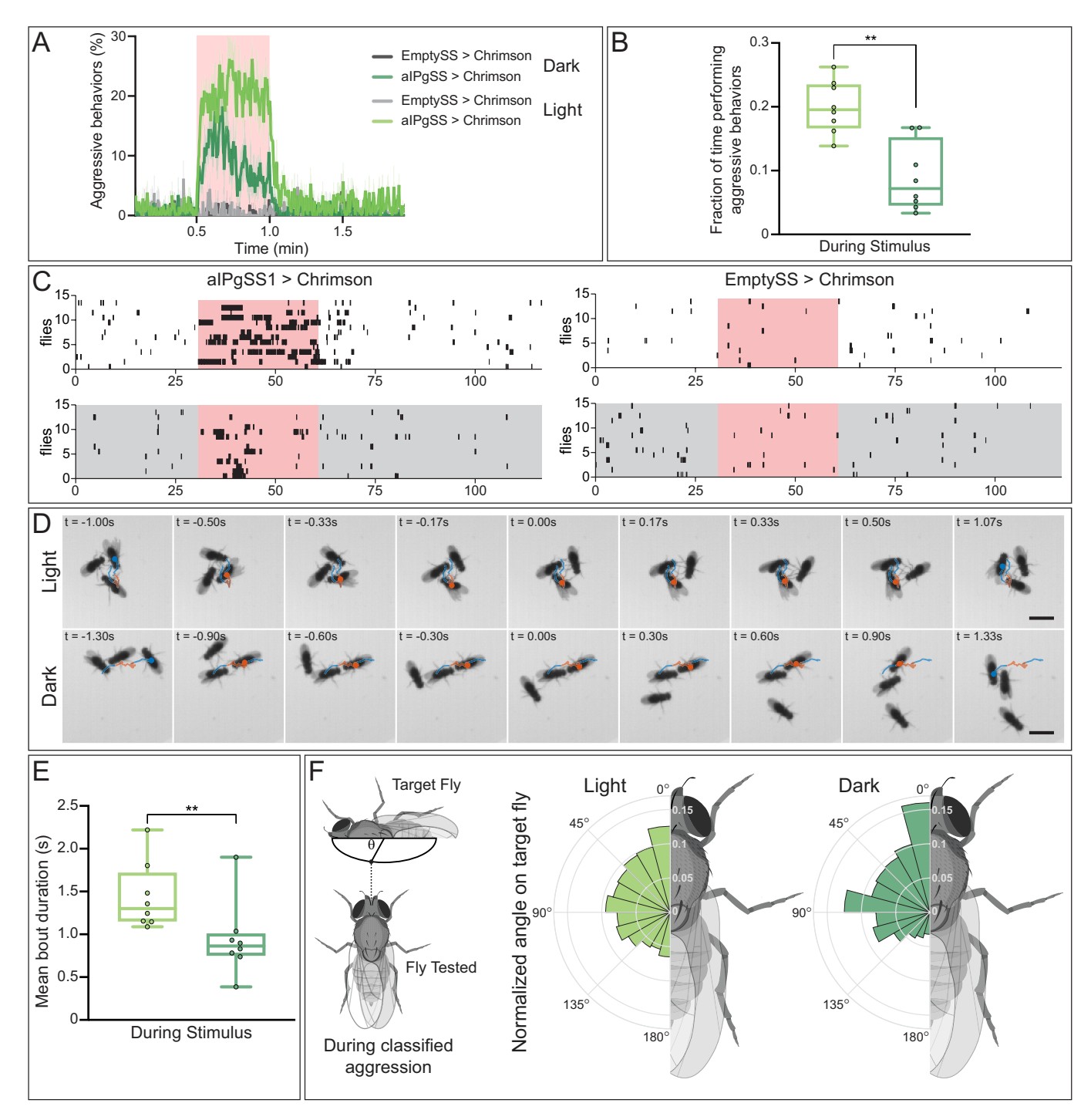

**Figure 9.** Absence of visual information alters aggressive behaviors during aIPgSS1 stimulation. (**A**) Percentage of individual female flies engaging in aggressive behaviors over the course of a 2 min trial during which a 30 s 0.1 mW/mm² continuous light stimulus was delivered (pink shading). Trials were conducted either in the presence or absence of visible light, as indicated. (**B**) Fraction of time individuals spent performing aggressive behaviors during the stimulus period. Each dot represents the mean of the individuals within each experiment. (**C**) Ethogram plots of the aggressive events in light (white) and dark (gray) conditions. Each row represents one individual in a trial and black bars represent classified aggressive events. The stimulus period is indicated in pink and the ethograms were chosen to represent a typical trial for each condition. (**D**) Time course of example aggressive events during a single bout in light (top) and dark (bottom) conditions. The trajectory of one fly is shown both prior to (blue), during (orange), and immediately following (blue) an aggressive bout. The time before (-) and after (+) a central frame of the behavior is shown in the upper left corner of each image. Note that although average bout length in longer in the light, we selected bouts of similar length for display to facilitate comparison. The scale bar (3

*Figure 9 continued on next page*

*Figure 9 continued*

mm) applies throughout each condition. See *Video 15* for examples of multiple bouts in the light and dark. (**E**) Mean length of classified aggressive behavioral events during the stimulus period. Event lengths were calculated from the per-frame classifications. Each dot represents the mean of the individuals within each experiment. (**F**) Polar plots of the angle on the targeted fly (θ), measured as shown in the diagram at the left, during classified aggressive events. Each wedge represents 15 degrees and plots were normalized within each condition. Light: EmptySS > Chrimson, n = 89 flies in eight experiments; aIPgSS1 > Chrimson, n = 136 flies in eight experiments; Dark: EmptySS > Chrimson, n = 90 flies in eight experiments; aIPgSS1 > Chrimson, n = 126 flies in eight experiments. Box-and-whisker plots in panels B and E show median and IQR; whiskers show range. A Mann-Whitney *U* test was used for statistical analysis. Asterisk indicates significance: \*\*p<0.01.

The online version of this article includes the following source data and figure supplement(s) for figure 9:

**Source data 1.** Source data for behavioural experiments in *Figure 9*.
**Figure supplement 1.** Connections between visual projection neurons, interneurons, and aIPg type 1, type 2 and type 3.

---

aIPg cell types—suggests that the aIPgs might influence many different circuits within the brain to promote aggression as well as likely influence behaviors not related to aggression. Examining the roles of aIPg's downstream targets provides an approach to understand how aIPg neurons exert their effects. However, most of these downstream targets have not been functionally characterized and genetic tools to manipulate them do not yet exist. Therefore, their identification does not immediately provide insight. Nevertheless, we did find that a substantial subset of aIPg's output is directed to the anterior optic tubercle (AOTU), where it is poised to gate the flow of visual information, revealing one way in which aIPg activation is likely to affect behavior.

## Absence of visual information alters aggressive behaviors following aIPg activation

As a simple approach to assess the role of visual information, we compared the results of aIPg activation in flies in a lighted versus a dark arena. Aggression was reduced in the absence of visible light (*Figure 9A,B*). Moreover, the structure of the behavior also differed when visual information was not available and there were fewer long duration bouts (*Figure 9C–E*; *Video 15*). A lack of visual information may alter the way in which individuals interact with one another during an aggressive bout. In addition to a shift towards shorter bout durations, flies in the darkened arena appear to have a stronger preference for targeting the head of the other fly than in the lighted arena (*Figure 9F*; *Video 15*). These behavioral differences suggest visual information may alter the substructure of aggressive events. From our connectomics analysis, we did indeed find input from visual projection neurons that is indirectly delivered to aIPg type 1–3 neurons (*Figure 9—figure supplement 1*). However, as described in the next section, we also discovered a pathway suggesting that aIPg activation may influence the way visual information is used by gating its flow through the AOTU, a brain region strongly targeted by aIPg type 1 and type 2 neurons.

## aIPg type 1 and type 2 neurons synapse onto projection neurons conveying visual information to motor pathways

We found strong connections from aIPg neurons to over ten cell types that provide direct pathways to descending interneurons (DNs), a neuronal population known to drive motor behavior (*Figure 10A*). Among these were several projection neurons that connect the AOTU, a target of many visual projection neurons from the optic lobe, to DNs. Six of these projection neurons received more than 90 synapses from the combination of aIPg type 1 and 2 neurons and then made over 100 synapses onto at least one of five different DN cell types. One of these strongly connected DNs, DNa02, has been recently implicated in steering behavior (*Rayshubskiy et al., 2020*). Since less than half of the known DNs (*Hsu and Bhandawat, 2016*; *Cande et al., 2018*; *Namiki et al., 2018*) have been identified to date in the hemibrain v1.1 dataset, the number of neurons we identified as participating in this circuit motif is almost certainly an underestimate. Two other projection-neuron cell-types connect aIPg neurons and LC10s with DNs with weaker strength connections (LAL026 and LAL027). In addition, SMP148, VES041, LAL025 and PS002 get only 0, 2, 4, and six percent, respectively, of their total synaptic input coming from identified visual projection neurons, raising the possibility that aIPg neurons also gate non-visual information (*Figure 10A*). We also found one case of a direct connection from aIPg type 3 to a DN as well as an indirect connection between pC1d and a DN (*Figure 10A*). Many of the connections from the aIPgs to DNs are ipsilateral, while AOTU019

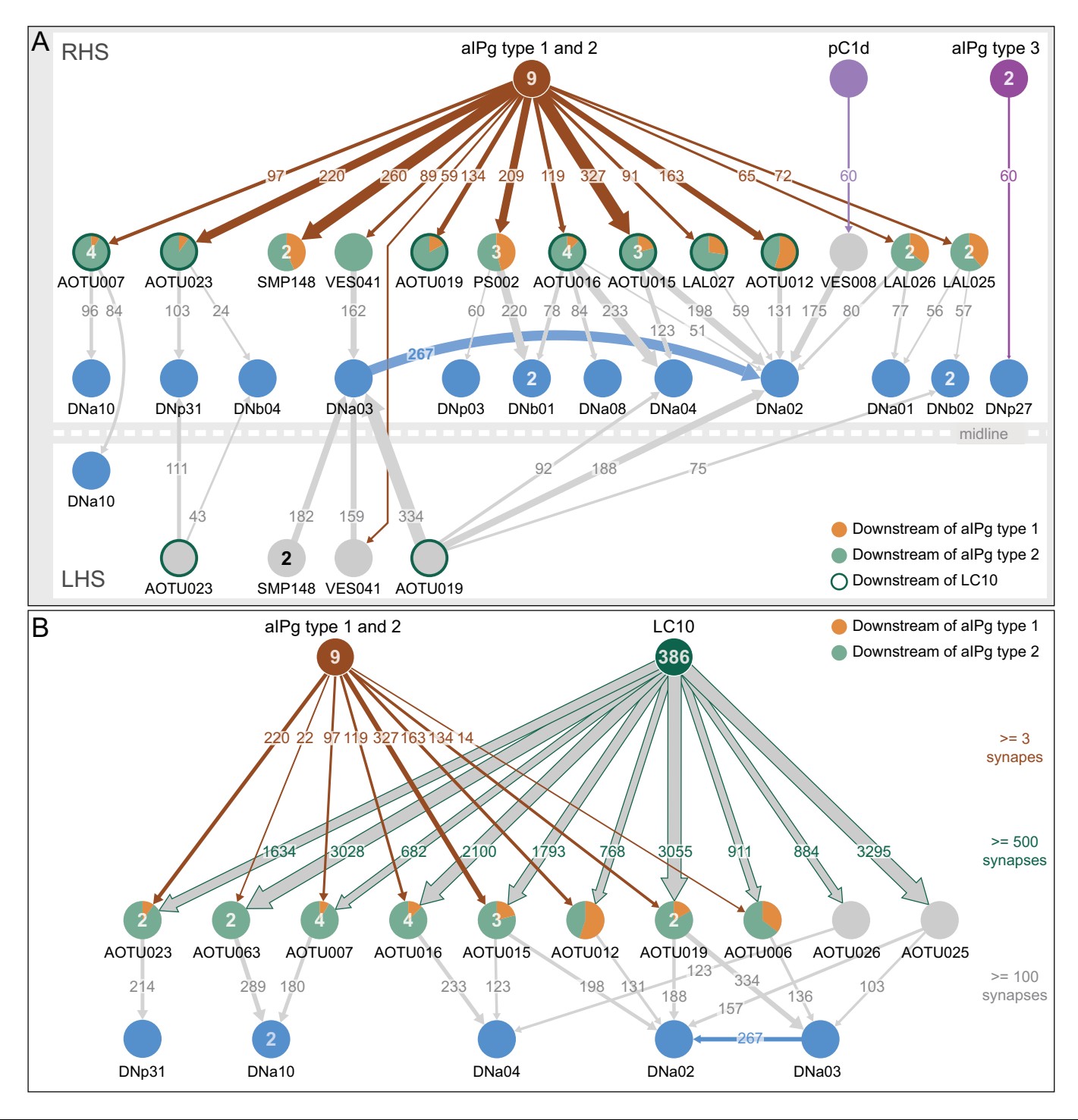

**Figure 10.** Connection paths of aIPg and pC1d neurons to descending neurons. (**A**) The numbers within the arrows represent synapse numbers; only connections with greater than 50 synapses between cell types are shown. Cell types are labelled and the number of neurons for cell types with more than one cell that participate in the diagrammed connections are given inside the circle representing the cell. The proportion of inputs from type 1 and type 2 aIPg neurons is indicated by the color-coding of the circle. AOTU-innervating neurons that also get input of more than 500 synapses from LC10 neurons are highlighted with a dark green circumference (see panel B). Some of the neurons downstream of aIPg (AOTU023 and VES041) make bilateral connections to DNs and others (AOTU019 and SMP148) make contralateral connections. For this reason, the right (RHS, above the line representing the midline) and left (LHS, below the line representing the midline) brain hemispheres are diagrammed separately to show all connections to right-hemisphere DNs. DNa03 is strongly connected to DNa02 in the LAL and VES. (**B**) AOTU-innervating cell types that make more than 100 synapses onto

*Figure 10 continued on next page*

*Figure 10 continued*

one or more DN cell types and that also get strong input (more than 500 combined synapses) from LC10 neurons are shown. The aIPg type 1 and type 2 cell types make inputs onto the majority of these AOTU neurons, as shown; the relative proportion of inputs from type 1 and type two neurons is indicated by color-coding of the circle representing the AOTU neuron. The numbers within the arrows represent synapse number. The numbers in the circles indicate the number of cells within that cell type in the right brain hemisphere that participate in the indicated connections. For example, 386 of the 449 LC10 cells identified in the hemibrain volume make connections to at least one of the shown AOTU neurons. Only about half the known DN cell types have been definitively identified in the hemibrain v1.1 dataset and thus there are likely to be additional DNs that meet our criteria that are not shown here.

The online version of this article includes the following figure supplement(s) for figure 10:

**Figure supplement 1.** Morphology and connectivity of AOTU019.

**Figure supplement 2.** Synaptic inputs from LC10 and aIPg neurons are segregated in different dendritic arbors of AOTU neurons.

and SMP148 make contralateral connections (*Figure 10A*, *Figure 10—figure supplements 1* and *2*). These connection patterns suggest involvement in promoting directed movement. In contrast, AOTU023 and VES041 make bilateral connections. It is also worth noting that AOTU019 has an axon of similar diameter to that of the giant fiber (*Figure 10—figure supplement 1*), implying that it might be capable of rapid conduction with high precision.

We next explored the connections between aIPg neurons, the AOTU neurons they innervate, and visual inputs to those AOTU neurons. We found that the six AOTU types that are innervated by aIPg neurons and connected to DNs, as described above, were also innervated by LC10 neurons, with a threshold of 500 synapses (*Figure 10B*). These five cell types account for nearly 10% of LC10's output (*Figure 12—figure supplement 1*). LC10 neurons can promote directional movement and reaching (*Wu et al., 2016*) and play a key role in small object detection and directed movement during courtship (*Ribeiro et al., 2018*; *Sten et al., 2020*). The synapses from LC10 and aIPg neurons were spatially segregated, with LC10 synapses largely confined to the AOTU and two thirds of those from the aIPg neurons occurring on a separate arbor in the SIP (*Figure 10—figure supplement 2*). Both aIPg and LC10 neurons (*Davis et al., 2020*) are cholinergic and may therefore cooperate in exciting their shared AOTU targets. It is striking that eight of the ten AOTU cell types that serve as major pathways connecting LC10 input to DNs also receive input from aIPg neurons (*Figure 10B*; *Figure 12—figure supplement 1A*). This circuit motif implies that an important role for aIPg neuronal activity in gating the flow of visual information to influence motor outputs.

## Discussion

The circuits that govern aggression in *Drosophila* are known to be sexually dimorphic and are poorly understood in females. In this paper, we described female aggressive behaviors, uncovered key components of the underlying neuronal circuits, developed genetic reagents to manipulate these neurons, and mapped their connections using EM-level connectomics. Specifically, we discovered the involvement of a subset of the aIP-g lineage, a collection of cell types not previously implicated in social behaviors, in mediating female aggressive social interactions. Optogenetic activation dramatically increased aggression in lines labelling aIPg neurons, while inactivation diminished these actions. Analysis using EM-level connectomics revealed strong connectivity between these aIPg neurons and two members of the pC1 cluster, a group of related cell types previously linked to social behaviors (*Hoopfer et al., 2015*; *Palavicino-Maggio et al., 2019*; *Wang et al., 2020a*; *Zhou et al., 2014*). In particular, pC1d is the top pre-synaptic input to aIPg neurons, devoting ~13% of its output synapses to them. Behavioral tests using split-GAL4 lines that cleanly label pC1d demonstrated its ability to increase female aggression. In contrast, we found no evidence for the involvement of the other four pC1 cell types, pC1a - c and pC1e, in promoting aggression. Consistent with these behavioral results, we found that three of these pC1 neurons, pC1a - c, were not connected to the aIPg neurons. As pC1e has significant connectivity with aIPg neurons, it may well play a role not revealed by our assays. Finally, we characterized the upstream inputs and downstream targets of aIPg and pC1d neurons. In the process, we identified multiple paths from aIPg neurons to DNs as well as a circuit motif by which aIPg neurons might increase the salience of visual information pertinent for aggressive behaviors.

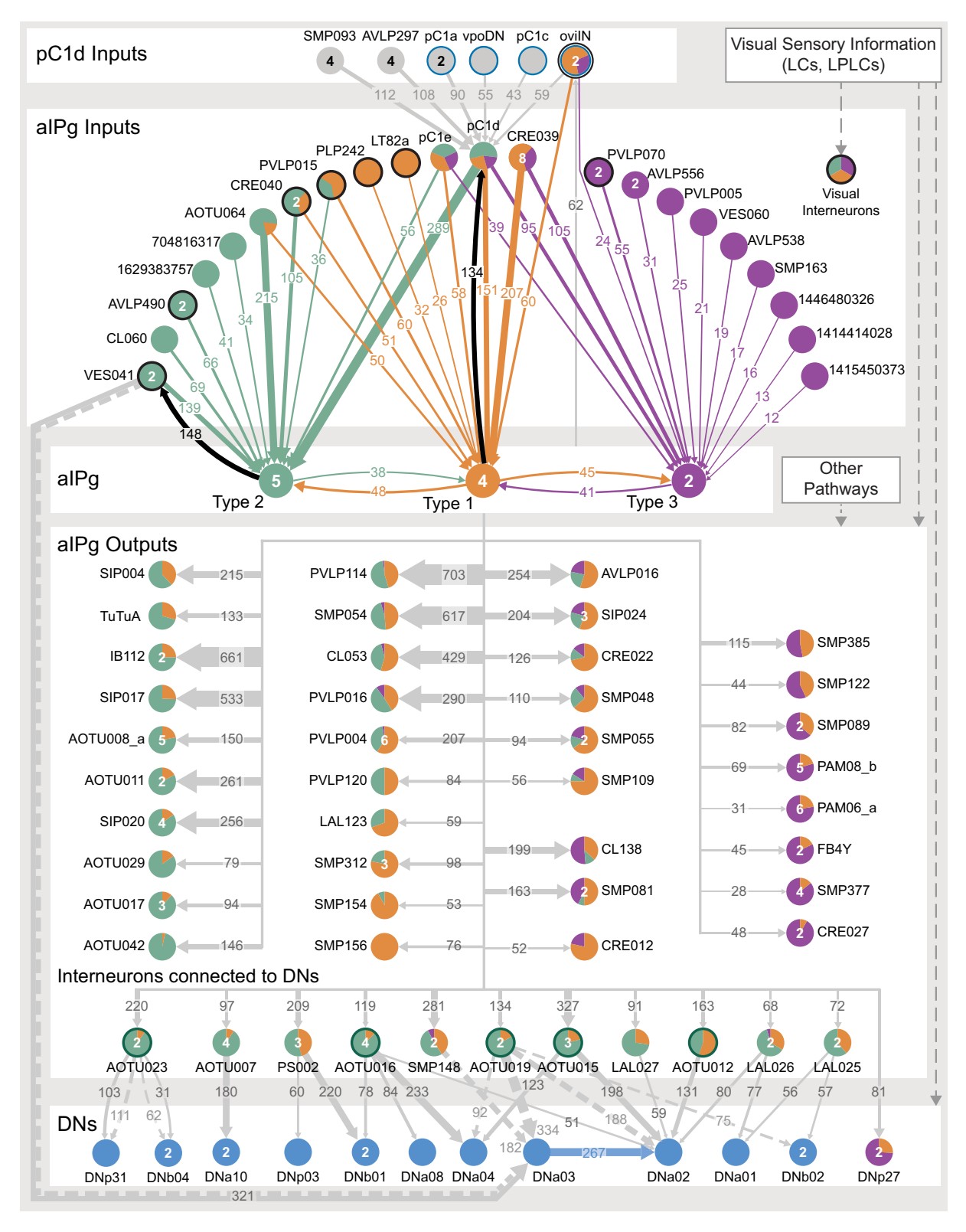

**Figure 11.** Proposed circuit underlying female aggressive behaviors. A diagram of key neuronal pathways centered on pC1d and aIPg type1, type 2 and type 3 neurons is shown. The numbers in arrows represent synapse number. We propose that pC1d facilitates female aggressive behaviors by acting through aIPg type 1 (orange), type 2 (green), and type 3 (purple) neurons. The top four inputs to pC1d and three other neurons of interest are shown. These include neurons implicated in other female behaviors (highlighted with a blue circumference), including oviposition (oviIN, pC1a) and

*Figure 11 continued on next page*

*Figure 11 continued*

mating (vpoDN), suggesting communication between the circuits underlying these behaviors and aggression. We propose that aIPg neurons integrate diverse upstream signals and then alter brain state in a way that increases the likelihood of engaging in aggressive social interactions. Upstream neuronal populations are color-coded by the proportion of their synapses they make onto the aIPg type 1, aIPg type 2, and aIPg type 3 subtypes. The post-synaptic targets of aIPg type 1, aIPg type 2, and aIPg type 3 neurons are shown color-coded by the relative proportion of their synaptic input they receive from each aIPg type. AOTU neurons that also get input from LC10 neurons (see *Figure 10B*) are highlighted with a dark green circumference. Inputs to the aIPg neurons were thresholded in proportion to the number of cells in each aIPg type resulting in thresholds of 24, 30, and 12 synapses to type 1, type 2, and type 3, respectively. Downstream outputs receiving more than 50 synapses from aIPg neurons (outputs from all 11 aIPg neurons pooled) are shown, with neurons receiving fewer synapses included when a majority came from aIPg type 3. For cell types that get input from multiple types of aIPg neurons, the relative proportions of inputs from different types are indicated by the color-coding of the circle. Connections with DNs with more than 50 synapses are shown. VES041 neurons have strong reciprocal connections with aIPg type 2 neurons; both the right and left hemisphere VES041 neurons are connected to the right-hemisphere DNa03 (321 total synapses; see *Figure 9A*) and are shown as solid and dotted arrows, respectively, in this diagram. Connections from AOTU019, SMP148, and AOTU023 from the contralateral hemisphere to DNs are similarly shown as dotted arrows (see *Figure 10A*).

## aIPg neurons mediate female aggressive behaviors

Innate behaviors, including aggression, have been proposed to result from the interplay of external stimuli and the internal state of the animal (*Lorenz, 1963*; *Tinbergen, 1951*). Neuronal populations that have the ability, when experimentally activated, to bypass normally required sensory cues to induce aggressive behavior have been previously identified. For example, male-specific neurons expressing *Drosophila* tachykinin (DTK) act as a hub mediating aggressive behaviors when activated and have been proposed to encode higher levels of motivation (*Anderson, 2016*; *Asahina, 2017*; *Asahina et al., 2014*; *Hashikawa et al., 2018*; *Hashikawa et al., 2017*; *Hoopfer, 2016*). As aIPg activity appears to be both necessary and sufficient to perform a high level of female aggression, aIPg neurons fit the definition of a mediator as proposed by *Gregg, 2003*.

The aIPg neurons in our split-GAL4 lines comprise three distinct cell types that differ in morphology and connectivity. Our efforts to derive lines specific for each of these cell types have been unsuccessful, which has limited our ability to explore their individual roles. Nor have we yet performed physiological experiments that might reveal distinct features of their responses when the circuit is activated. As there are many differences in the inputs and outputs between the aIPg types (*Figure 11*), these three cell types are likely to make distinct contributions to the phenotypes we observed.

## Involvement of pC1d in female aggression and connections to neurons implicated in other female social behaviors

Neurons that provide input to the aIPgs could act as facilitators adjusting the degree of behavior or conveying specific sensory information important for its initiation and execution. For example, pC1d activation was able to drive aggressive behaviors but was not essential for them; rather it appears to be a focal point for information about other social behaviors. Stimulation of pC1e neurons alone did not result in aggressive phenotypes under the conditions tested, or increase the aggression observed when activated with pC1d, despite being a significant synaptic input to aIPg neurons. Additional experiments with other behavioral assays will be required to explore pC1e's functions.

A complementary study by *Deutsch et al., 2020* found that combined activation of pC1d and pC1e using ReaChR resulted in a persistent phenotype lasting for minutes. Using lines labeling aIPg cells, we did observe head butting and fencing behaviors after the cessation of the optogenetic stimulus that lasted for a shorter, but ethologically relevant (*Bath et al., 2017*; *Yuan et al., 2014*), time period on the order of tens of seconds. In our experiments, we did not observe similar persistence with pC1d under the majority of conditions tested. There are several experimental differences between the two studies that might explain our findings regarding the extent of persistence, including: the effector, stimulus duration, magnitude and wavelength of light, temporal pattern of stimulus, housing and testing conditions, as well as the media used. Consistent with this speculation, *Deutsch et al., 2020* show that shorter stimulation protocols led to less persistent behavior. Additionally, both our and *Deutsch et al., 2020*'s connectomic analyses documented reciprocal connectivity between pC1d and aIPg neurons. Feedback between aIPg type 1 neurons and pC1d may play

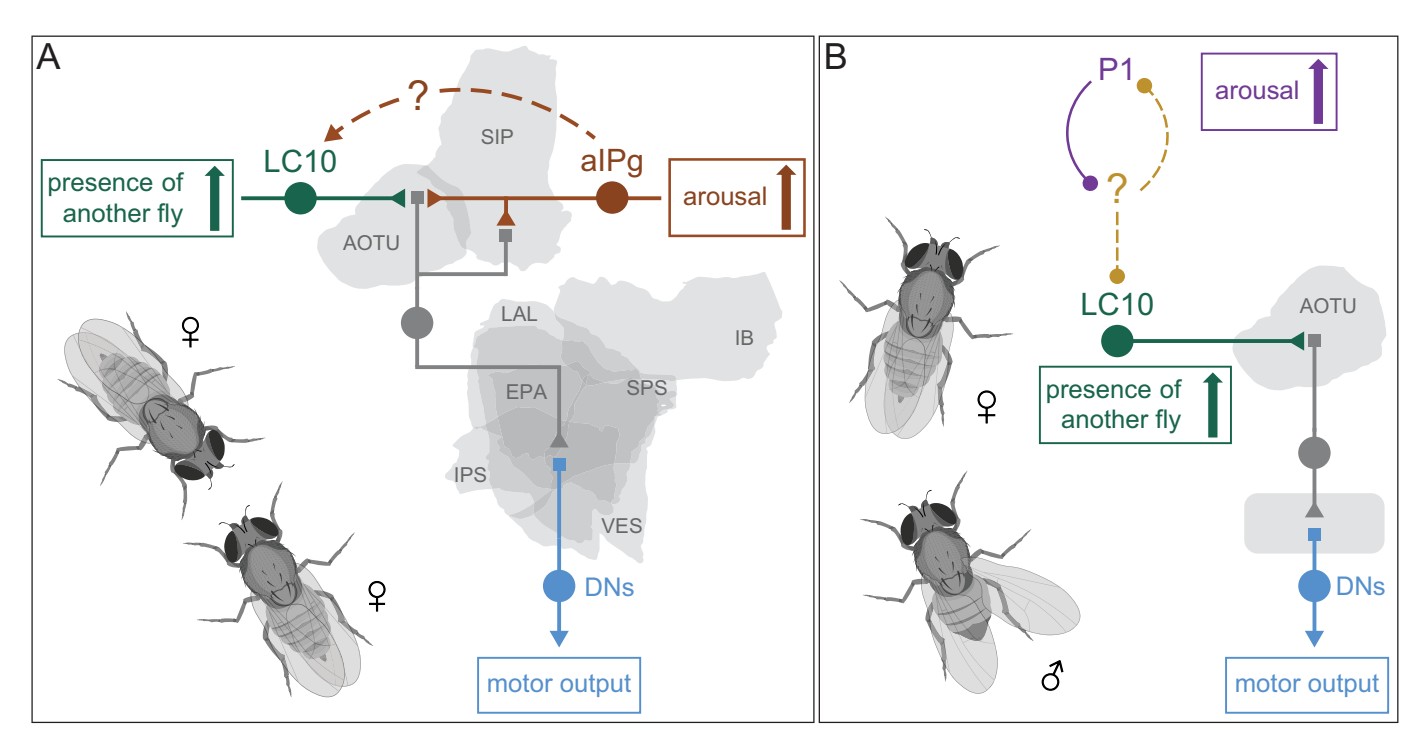

**Figure 12.** Alternative models for how the saliency of visual input from LC10 neurons is increased by arousal during two different social behaviors. (**A**) A conceptual model for gating of visual information by aIPg neurons in females during aggression. Several types of projection neurons (represented by the gray neuron) connect the AOTU with ventral brain regions where they then innervate descending neurons (blue) to drive motor behavior (*Figure 10B*). These projection neurons get input from both aIPg neurons and LC10 neurons, a population of visual projection neurons that are tuned to be activated by a nearby moving fly. The combined activation of these excitatory cholinergic inputs might act synergistically to drive AOTU firing and signaling to DNs. At this point we do not know if the anatomical separation of the aIPg and LC10 inputs to the AOTU neurons (*Figure 10—figure supplement 2*) is simply a wiring convenience or has functional significance. The dotted arrow from aIPg to LC10 indicates an indirect path mediated by two types of TuTuA neurons (*Figure 12—figure supplement 1*). (**B**) A conceptual model based on the work of *Sten et al., 2020* for the gating of visual information by P1 neurons in males during courtship. P1 neurons are proposed to act upstream of LC10 such that LC10 neurons produce more synaptic output when male flies are aroused. The nature of the signal transmission between P1 and LC10 has not been determined. See *Sten et al., 2020* for details. We note that in addition to transmission at chemical synapses, signaling by neuropeptides also remains a possibility in both models: RNA profiling of LC10a and LC10b neurons from a mixture of males and females (*Davis et al., 2020*) has shown that both these cell types express the receptor for sNPF (sNPF-R) and our work showed that aIPg neurons express sNPF (*Figure 1—figure supplement 5*). Similarly, LC10a neurons express two receptors for Tk (TkR86C and TkR99D), while LC10b neurons appear to express only TkR86C at high levels. Tk is known to be an important regulator of male aggression (*Asahina et al., 2014*), but a role for Tk in courtship has not been established.

The online version of this article includes the following figure supplement(s) for figure 12:

**Figure supplement 1.** Connectivity of LC10 neurons in the AOTU.

a role in the extension of aggressive behavior during the off period following aIPg activation, a possibility investigated further in *Deutsch et al., 2020*.

Analysis of inputs to pC1d revealed links with neurons involved in female-specific and other social behaviors, including oviposition and courtship. Links between oviposition and female aggression have been observed in field work on other *Drosophila* species, with increased aggressive behaviors occurring on egg laying sites (*Shelly, 1999*). Supporting this close relationship between the behaviors, pC1d receives innervation from pC1a, a cell type involved in the egg laying pathway (*Wang et al., 2020a*). Interestingly, *Wang et al., 2020a* also noted differences in the depolarization of pC1d and pC1e following activation of the sex-peptide abdominal ganglion neuron. While these are not as extensive as seen in pC1a, they may provide a neural basis for the reported effect of mating status on female aggressive behavior (*Bath et al., 2020*; *Bath et al., 2018*; *Ueda and Kidokoro, 2002*). Further supporting the close tie between these two social behaviors, we found that nearly a quarter of the synaptic output in the central brain of vpoDN, a descending neuron involved in female

receptivity (*Wang et al., 2020b*) goes to pC1d. Additionally, an inhibitory neuron implicated in modulating oviposition and mating behavior, oviIN, innervates both pC1d (*Wang et al., 2020a*; *Wang et al., 2020b*) and aIPg neurons. Mating behavior and aggression also overlap at the P1/pC1 level in male flies, raising the possibility of a common—but sexually dimorphic—node in both sexes (*Hoopfer et al., 2015*; *Wang et al., 2020a*). Although pC1d activation elicited aggressive behaviors, neither we nor *Palavicino-Maggio et al., 2019* were able to demonstrate its requirement for the execution of aggression.

## Activation of aIPg neurons results in a change to the individual's state

During social encounters, the decision to engage in mating, aggressive, defensive or other behaviors is influenced in part by the internal state of the organism, which can include motivation and arousal (*Asahina, 2018*; *Berridge, 2004*; *LeDoux, 2012*; *Tinbergen, 1951*; *Lorenz and Leyhausen, 1973*). However, how these states are encoded within the brain and subsequently influence behavior is not well understood. Does aIPg act by changing brain state? The activation of aIPg neurons does not produce an immediate, synchronous response in all the individual flies in an arena. This contrasts with the activation of the lobula columnar cell type LC16 under the same experimental conditions (*Wu et al., 2016*), which results in all the flies immediately walking backward, a type of response associated with command neurons. Our results support the view that, rather than acting as a command neuron, aIPg neurons act by modifying the way in which a fly responds to external stimuli such that its activation increases the likelihood that an individual will engage in aggressive behaviors.

Social interactions are influenced by environmental stimuli in addition to the internal state of the individual (*Chen and Hong, 2018*). Our experiments on aggressive behavior in female flies were consistent with prior studies demonstrating the importance of food availability, proximity, and prior social interactions (*Bath et al., 2017*; *Ueda and Kidokoro, 2002*). As aIPg activation resulted in the execution of aggressive behaviors independent of these features, our results suggest that aIPg acts downstream, or in the integration, of these sensory inputs.

How do we envision aIPg-induced changes in internal state affect behavior? One way might be to alter the saliency of particular streams of sensory information. Our anatomical analyses uncovered a circuit motif that appears to be designed to do just that for visual information conveyed by LC10 neurons, a population of visual projection neurons well-tuned for tracking other flies during social interactions (*Ribeiro et al., 2018*; *Sten et al., 2020*). More specifically, we found that aIPg and LC10 have shared targets in the AOTU, projection neurons connected to DNs that drive motor actions. We propose that these projection neurons integrate excitatory signals from aIPg and LC10 neurons (*Figure 12A*).

Activation of aIPg neurons may also modify LC10-mediated visual information in other ways. Our connectomic analysis revealed potential pathways by which aIPg neurons might act further upstream to adjust LC10's activity (*Figure 12*, *Figure 12—figure supplement 1*). Recent work by *Sten et al., 2020* proposed that the activity of P1 neurons in males changes the salience of visual cues from LC10 during courtship behaviors. While the circuit mechanisms in males have not yet been uncovered, it is clear from *Sten et al., 2020* that P1 exerts its influence upstream of LC10's outputs (*Figure 12B*). P1 is more analogous to the pC1 component of the circuit, as both neurons are members of the sexually dimorphic P1/pC1 cluster (*Asahina, 2018*). It is currently unclear, in the absence of connectomic information, whether males have a cell type that occupies a circuit position analogous to that of aIPg neurons. It will be interesting to compare the mechanisms being used to increase the salience of the same population of LC10 neurons during different social behaviors, as well as to explore how these pathways influence the progression of aggressive behaviors.

## Concluding remarks

The female fly must integrate information from sensory cues about the external world—such as the presence of other individuals, food and egg laying sites—with internal states such as hunger and arousal—when weighing whether to initiate or abandon aggressive interactions. Our work provides a foundation for future studies aimed at understanding the complex social behavior of aggression by identifying key neuronal cell types, placing them in the context of a larger neuronal circuits, and providing tools for their manipulation.

# Materials and methods

**Key resources table**

| Reagent type (species) or resource | Designation | Source or reference | Identifiers | Additional information |
|---|---|---|---|---|
| Genetic reagent (*Drosophila melanogaster*) | VT064565-x-VT043699 | this paper | splitgal4.janelia.org: SS36564 | split-GAL4 driver line targeting aIPg (aIPgSS1) |
| Genetic reagent (*Drosophila melanogaster*) | VT043699-x-R11A07 | this paper | splitgal4.janelia.org: SS36551 | split-GAL4 driver line targeting aIPg (aIPgSS2) |
| Genetic reagent (*Drosophila melanogaster*) | R11A07-x-VT043700 | this paper | splitgal4.janelia.org: SS32237 | split-GAL4 driver line targeting aIPg (aIPgSS3) |
| Genetic reagent (*Drosophila melanogaster*) | R72C11-x-R11A07 | this paper | splitgal4.janelia.org: SS47478 | split-GAL4 driver line targeting aIPg (aIPgSS4) |
| Genetic reagent (*Drosophila melanogaster*) | R11A07-x-R72C11 | this paper | splitgal4.janelia.org: SS56964 | split-GAL4 driver line targeting aIPg (aIPgSS5) |
| Genetic reagent (*Drosophila melanogaster*) | R35C10-x-R71A09 | this paper | splitgal4.janelia.org: SS56987 | split-GAL4 driver line targeting pC1d (pC1dSS1) |
| Genetic reagent (*Drosophila melanogaster*) | R71A09-x-R26F09 | this paper | splitgal4.janelia.org: SS57598 | split-GAL4 driver line targeting pC1d (pC1dSS2) |
| Genetic reagent (*Drosophila melanogaster*) | VT25602-x-VT002064 | this paper | splitgal4.janelia.org: SS43274 | split-GAL4 driver line targeting pC1d and pC1e (pC1dSS3) |
| Genetic reagent (*Drosophila melanogaster*) | R35C10-x-VT002063 | this paper | splitgal4.janelia.org: SS59331 | split-GAL4 driver line targeting pC1d and pC1e (pC1dSS4) |
| Genetic reagent (*Drosophila melanogaster*) | R35C10-x-R60G08 | this paper | splitgal4.janelia.org: SS59336 | split-GAL4 driver line targeting pC1e (pC1eSS1) |
| Genetic reagent (*Drosophila melanogaster*) | R60G08-x-VT002063 | this paper | splitgal4.janelia.org: SS39313 | split-GAL4 driver line targeting pC1e (pC1eSS2) |
| Genetic reagent (*Drosophila melanogaster*) | R60G08-x-R35C10 | this paper | splitgal4.janelia.org: SS59433 | split-GAL4 driver line targeting pC1e (pC1eSS3) |
| Genetic reagent (*Drosophila melanogaster*) | R18A05-x-dsx-ZpGal4DBD | this paper; gift of K. Wang and B. Dickson | SS75230 | split-GAL4 driver line targeting pC1a-c |
| Genetic reagent (*Drosophila melanogaster*) | BPp65AD-x-BPZpGal4DBD | *Pfeiffer et al., 2010*; *Aso et al., 2014* | emptySS | split-GAL4 driver line empty brain control |
| Genetic reagent (*Drosophila melanogaster*) | 20XUAS-CsChrimson-mVenus | *Klapoetke et al., 2014*; *Aso et al., 2014* | | |
| Genetic reagent (*Drosophila melanogaster*) | 10XUAS-CsChrimson-mVenus | *Klapoetke et al., 2014*; *Aso et al., 2014* | | |
| Genetic reagent (*Drosophila melanogaster*) | 5XUAS-CsChrimson-mVenus | *Klapoetke et al., 2014*; *Aso et al., 2014* | | |
| Genetic reagent (*Drosophila melanogaster*) | MCFO-1 | *Nern et al., 2015* | | |
| Genetic reagent (*Drosophila melanogaster*) | pJFRC2-10XUAS-IVS-mCD8::GFP | *Pfeiffer et al., 2010* | | |
| Genetic reagent (*Drosophila melanogaster*) | UAS-impTNT-HA | *Sweeney et al., 1995* | | |
| Genetic reagent (*Drosophila melanogaster*) | UAS-TNT-E | *Sweeney et al., 1995* | | |
| Antibody | nc82 supernatant; Mouse α-bruchpilot monoclonal | Developmental Studies Hybridoma Bank | Cat #: nc82-s | https://www.janelia.org/project-team/flylight/protocols Dilution, (1:30) |
| Antibody | Mouse α-V5 Tag monoclonal | AbD Serotec | Cat #: MCA1360D550GA | Dilution (1:500) |
| Antibody | Rabbit α-HA Tag monoclonal | Cell Signaling Technologies | Cat #: 3724S | Dilution (1:300) |
| Antibody | Rat α-FLAG Tag (DYKDDDDK Epitope tag) monoclonal | Novus Biologicals | Cat #: NBP1-06712 | Dilution (1:200) |

*Continued on next page*

*Continued*

| Reagent type (species) or resource | Designation | Source or reference | Identifiers | Additional information |
|---|---|---|---|---|
| Antibody | rabbit anti-GFP polyclonal | Thermo Fisher Scientific | Cat #: A-11122; RRID:AB_221569 | Dilution, (1:1000) |
| Antibody | Alexa Fluor 488 goat anti-rabbit polyclonal | Thermo Fisher Scientific | Cat #: A-11034; RRID:AB_2576217 | Dilution, (1:800) |
| Antibody | Alexa Fluor 568 goat anti-mouse polyclonal | Thermo Fisher Scientific | Cat #: A-11031; RRID:AB_144696 | Dilution, (1:400) |
| Antibody | ATTO 647N goat anti-rat polycolonal | Rockland Immunochemicals Inc | Cat #: AB_10893386 | Dilution, (1:300) |
| Chemical compound, drug | paraformaldehyde | Electron Microscopy Sciences | 15713 s | https://www.janelia.org/project-team/flylight/protocols |
| Chemical compound, drug | Triton X-100 | Sigma–Aldrich | X100 | https://www.janelia.org/project-team/flylight/protocols |
| Chemical compound, drug | DPX Mountant | Electron Microscopy Sciences | #13512 | https://www.janelia.org/project-team/flylight/protocols |
| Chemical compound, drug | xylene | Fisher Scientific | x5-500 | https://www.janelia.org/project-team/flylight/protocols |

## Fly stocks

All experiments used virgin female flies unless otherwise stated. Flies were reared on standard corn-meal molasses food at 25°C and 50% humidity. For optogenetic activation experiments, flies were reared in the dark on standard food supplemented with retinal (Sigma-Aldrich, St. Louis, MO) unless otherwise specified, 0.2 mM all *trans*-retinal prior to eclosion and 0.4 mM all *trans*-retinal post eclosion. Hemidriver lines were created using gateway cloning as previously described (*Dionne et al., 2018*).

## Optogenetic activation behavioral testing

Groups of 13–18 group-housed virgin female flies (5–10 days post-eclosion) were video recorded at 25°C and 50% relative humidity in a 127 mm circular arena with a center depth of 3.5 mm as described previously (*Robie et al., 2017*; *Wu et al., 2016*). All tests were conducted under visible light conditions at ZT0 to ZT4 unless otherwise stated. Flies were loaded into the arena using an aspirator. For activation of neurons expressing CsChrimson, the arena was illuminated as specified in the figure legends; unless otherwise stated we used constant uniform illumination with 617 nm LEDs (Red-Orange LUXEON Rebel LED - 122 lm; Luxeon Star LEDs, Brantford, Canada). Unless otherwise stated, all trials were performed under white-light illumination. For each trial, flies were acclimatized to the area for 30 s prior to the delivery of a single constant stimulus lasting 30 s. Pulse stimulation at 0.1 mW/mm$^2$ was given in 30 s intervals with an inter-stimulus interval of 30 to 60 s. The pulse width was kept constant at 10 ms while the pulse number and period varied. Videos were recorded from above using a camera (ROHS 1.3 MP B and W Flea3 USB 3.0 Camera; Point Gray, Richmond, Canada) with an 800 nm long pass filter (B and W filter; Schneider Optics, Hauppauge, NY) at 30 frames per second and 1024 × 1024 pixel resolution. Flies were tracked using Ctrax (*Branson et al., 2009*) followed by automated classification of behavior with JAABA classifiers (see *Kabra et al., 2013*). Previously validated classifiers for touch, chase, and walking were trained from videos using a similar behavioral assay and conditions (*Robie et al., 2017*). Additionally, we created a novel classifier for aggression which compassed both fencing and head butting. We validated the performance of this classifier against manually labeled ground truth data using videos that were not part of the training dataset (*Supplementary file 1*; 90.9% (true positive) and 96.1% (true negative) framewise performance). For figures displaying behavioral time courses, the mean of 0.5 s (15-frame) bins is shown. For experiments examining individual behaviors, the tracking of individual flies was manually reviewed and corrected when necessary using the FixErrors GUI available with Ctrax. Bout lengths in *Figure 9C* were calculated from the per-frame classifications during the stimulus period but were not truncated by the end of the stimulus period. In *Figure 9F*, the angle on targeted fly was

calculated as described for the 'angle on closest fly' per-frame feature in *Robie et al., 2017* with increased precision to 150 points evenly spaced along the circumference of the ellipse.

Activation experiments detailed in *Figure 2*, *Figure 2—figure supplements 1* and *2*, and *Figure 4—figure supplement 11* were performed in 16 mm diameter x 12 mm height chambers using same LEDs and camera setup as detailed above. For these experiments, two flies (genotypes and sex specified in figure) were introduced into the arena with an aspirator. In experiments with both activated and wild-type females, the wings of the wild-type flies were clipped three days prior to the experiment to allow the genotypes to be distinguished. Flies were acclimatized to the arena for 30 s prior to delivery of a single constant stimulus lasting 30 s. Flies were tracked using the Caltech Fly-Tracker (http://www.vision.caltech.edu/Tools/FlyTracker/) followed by automated classification of behavior with a JAABA classifier for head butting and fencing behaviors (see *Supplementary file 1*; 90.1% (true positive) and 95.3% (true negative) framewise performance). Calculations of the fraction of time spent performing a behavior were made using the score files and averaging over the period indicated.

For high-speed videos, two flies were loaded into an arena using cold anesthesia and an aspirator. The arena was illuminated with a ring light and a LED gooseneck. Constant stimulation was provided from above during the recording at 0.45 mW/mm$^2$ with a 625 nm light source. Videos were recorded with a Photon Fastcam Mini camera at 1000 frames per second.

Receptivity assays (*Figure 2—figure supplement 2*) were performed as detailed in *Wang et al., 2020a*. Briefly, one virgin female and one wild-type single housed virgin male were transferred into a 10 mm diameter x 2 mm height arena through aspiration. Flies were videotaped under white-light illumination for 15–30 min and photostimulated at 0.1 mW/mm$^2$. The copulation latency was scored manually.

## Inactivation behavioral testing

Virgin female flies were group-housed at a density of 20–40 females per vial at 22°C on dextrose media (79 g agar, 275 g yeast, 520 g cornmeal, 1100 g Dextrose, 87.5 mL 20% Tegosept, 20 mL Propionic Acid in 11000 mL of water) for 21–28 days. Female flies were single housed for 5–7 days and then starved on 1% agarose in water for the 20–24 hr immediately prior to testing. Rearing and housing were performed in a light cycling incubator set for 12/12 hr light/dark cycle. All inactivation experiments were run at 25°C and 40% humidity and performed during ZT0 to ZT3 with two female flies per arena.

Assays were performed in previously described acrylic multi-chamber aggression arenas (*Hoopfer et al., 2015*; *Kim et al., 2018*; *Asahina et al., 2014*). Each circular arena (16 mm diameter x 12 mm height) was coated with Insect-a-Slip (Bioquip Products, Rancho Dominguez, CA) on the walls and SurfaSil Siliconizing Fluid (Thermo Fisher Scientific, Waltham, MA) on the clear acrylic top plate to confine the flies to the bottom plate. The floor of the arenas was composed of a uniform layer (~1 mm thick) of apple juice-agarose food [2.5% (w/v) sucrose and 2.25% (w/v) agarose in apple juice] that contained a ~ 1 mm spot of live yeast (Fleischmann, Cincinnati, Ohio) placed in approximately the center of each arena. Flies were illuminated from beneath with visible light and recorded with ambient overhead room lighting. Flies were introduced into the chamber with gentle aspiration through a hole in the top plate and allowed to acclimatize for 30 s - 1 min prior to recording. For automated analysis, flies were tracked using the Caltech FlyTracker (http://www.vision.caltech.edu/Tools/FlyTracker/) followed by automated classification of behavior with a JAABA classifier for head butting and fencing behaviors (see *Supplementary file 1*; 87.0% (true positive) and 90.8% (true negative) framewise performance). For manual analysis, videos were blinded and scored for head butting and fencing behaviors using JWatcher software (http://www.jwatcher.ucla.edu/) and data were reported for each pair tested.

## Immunohistochemistry and imaging

Dissection and immunohistochemistry of fly brains were carried out as previously described (*Jenett et al., 2012*; *Aso et al., 2014*). Each split-GAL4 line was crossed to the same Chrimson effector used for behavioral analysis. Full step-by-step protocols can be found at https://www.janelia.org/project-team/flylight/protocols.

For single-cell labelling of aIPg neurons, we used the MultiColor FlpOut (MCFO) technique (*Nern et al., 2015*). For MultiColor FlpOut (MCFO) experiments, the MCFO stock was crossed to a split-GAL4 line. Flies were collected after eclosion, transferred to a new food vial and incubated in a 37°C water bath for 20–25 min. These flies were dissected and underwent whole-mount immunohistochemistry and confocal imaging (*Dolan et al., 2018*). The MCFO protocol from *Nern et al., 2015* was modified by the replacement of Alexa Fluor 647 goat anti-rat with ATTO 647N goat anti-rat at the same concentration.

All imaging was performed on LSM 700, 710 and 780 confocal microscopes (Zeiss) using ZEN software with a custom MultiTime macro. For all images except those in *Figure 1—figure supplement 1*, *Figure 4—figure supplement 1*, and *Figure 4—figure supplement 2*, the macro was programmed to automatically select appropriate laser power and/or gain for each sample, resulting in independent image parameters for each sample. The images in *Figure 1—figure supplement 1*, *Figure 4—figure supplement 1*, and *Figure 4—figure supplement 2* were captured on the same LSM 710 microscope with all imaging parameters held fixed to maximize comparability.

## RNA profiling of aIPg and pC1 cell types

Two different types of sequencing protocols were performed: bulk sequencing for aIPgSS3 and aIPgSS4 and low-cell sequencing for pC1dSS1, pC1dSS3, pC1eSS1, and pC1eSS3. Sequence data and additional details of the methods used for data processing can be found in the NCBI Gene Expression Omnibus database, accession number GSE158748.

### Expression checks

Neurons of interest were isolated by expressing the fluorescent reporter 10XUAS-IVS-myr::GFP; 10XUAS-IVS-nls-tdTomato using split-Gal4 drivers specific for particular cell types and then manually picking the fluorescent neurons from dissociated brain tissue. As a preliminary to the sorting process, each driver/reporter combination was examined to determine if the marked cells were sufficiently bright to be sorted effectively and that there was no off-target expression in neurons other than those of interest. Only drivers that met both these requirements were used.

### Dissociation of neurons

For each sample, 60–100 brains from 3 to 5 days post-eclosion adults were dissected in freshly prepared, ice cold Adult Hemolymph Solution (AHS, 108 mM NaCl, 5 mM KCl, 2 mM CaCl2, 8.2 mM MgCl2, 4 mM NaHCO3, 1 mM NaH2PO4, 5 mM HEPES, 6 mM Trehalose, 10 mM Sucrose), and the major tracheal branches removed. The brains were transferred to a 1.5 mL Eppendorf tube containing 500 µL of 1 mg/mL Liberase DH (Roche, prepared according to the manufacturer's recommendation) in AHS, and digested for 1 hr at room temperature. The Liberase solution was removed and the brains washed twice with 800 µL ice-cold AHS. The final wash was removed completely and 400 µL of AHS+2% Fetal Bovine Serum (FBS, Sigma) were added. The brain samples were gently triturated with a series of fire-polished, FBS-coated Pasteur pipettes of descending pore sizes until the tissue was homogenized, after which the tube was allowed to stand for 2–3 min so that the larger debris could settle.

### Sorting of labelled neurons

For hand sorting, the cell suspension was transferred to a Sylgard-lined Glass Bottom Dish (Willco Wells) and were allowed to settle for 10–30 min. Fluorescent cells were picked by aspiration using a pulled Kwik-Fil Borosilicate Glass capillary (Fisher), transferred to a Sylgard-lined 35 mm Mat Tek Glass Bottom Microwell Dishes (Mat Tek) filled with 170 µL AHS+2%FBS, allowed to settle, and then re-picked. Three washes were performed in this way before the purified cells were picked and transferred into 3 ul Lysis Buffer (0.2% Triton X-100, 0.1 U/µL RNAsin) for the low-cell method or, for the bulk RNA-seq method, into 50 µL extraction buffer from the PicoPure RNA Isolation Kit (Life Technologies) and then lysed for 5 m at 42°C and stored at −80°C.

For automated cell sorting, the samples were triturated in AHS+2%FBS that was run through a 0.2 µm filter, and the cell suspension was passed through a Falcon 5 mL round-bottom tube fitted with a 35 µm cell strainer cap (Fisher). Samples were sorted on a SONY SH800 cell sorter gated for single cells with a fluorescence intensity exceeding that of a non-fluorescent control. For bulk

samples, GFP+tdTom+ events were purity sorted directly into 50 microliters PicoPure extraction buffer, the sample lysed for 5 m at 42℃, and stored at −80℃. For low-cell samples, replicates of approximately 40 cells were sorted to enrich for GFP+tdTom+ cells into 3 μL Lysis Buffer and flash frozen on dry ice.

### Library preparation and sequencing

For bulk samples, total RNA was extracted from 400 to 800 pooled cells using the PicoPure kit (Life Technologies) according to the manufacturer's recommendation, including the on-column DNAse1 step. The extracted RNA was converted to cDNA and amplified with the Ovation RNA-Seq System V2 (NuGEN), and the yield quantified by NanoDrop (Thermo). The cDNA was fragmented, and the sequencing adaptors ligated onto the fragments using the Ovation Rapid Library System (NuGEN). Library quality and concentration was determined with the Kapa Illumina Library Quantification Kit (KK4854, Kapa Biosystems), and the libraries were pooled and sequenced on an Illumina NextSeq 550 with 75 base pair reads. Sequencing adapters were trimmed from the reads with Cutadapt v2.10 (*Martin, 2011*) prior to alignment with STAR v2.7.5c (*Dobin et al., 2013*) to the *Drosophila* r6.34 genome assembly (FlyBase). The resulting transcript alignments were passed to RSEM v1.3.0 (*Li and Dewey, 2011*) to generate gene expression counts.

For low-cell samples, one microliter of harsh lysis buffer (50 mM Tris pH 8.0, 5 mM EDTA pH 8.0, 10 mM DTT, 1% Tween-20, 1% Triton X-100, 0.1 g/L Proteinase K (Roche), 2.5 mM dNTPs (Takara), and ERCC Mix 1 (Thermo Fisher) diluted to 1e-7) and 1 μl10 μM barcoded RT primer were added, and the samples were incubated for 5 m at 50℃ to lyse the cells, followed by 20 m at 80℃ to inactivate the Proteinase K. Reverse transcription master mix (2 μL 5X RT Buffer [Thermo Fisher], 2 μL 5M Betaine [Sigma-Aldrich], 0.2 μL 50 μM E5V6NEXT template-switch oligo [Integrated DNA Technologies], 0.1 μL 200 U/μL Maxima H-RT [Thermo Fisher], 0.1 μL 40 U/μL RNAsin (Lucigen), and 0.6 μL nuclease-free water [Thermo Fisher]) was added to the lysis reaction and incubated at 42℃ for 1.5 hr, followed by 10 m at 75℃ to inactivate the reverse transcriptase. PCR was performed by adding 10 μL 2X HiFi PCR Mix (Kapa Biosystems) and 0.5 μL 60 μM SINGV6 primer and incubating at 98℃ for 3 m, followed by 20 cycles of 98℃ for 20 s, 64℃ for 15 s, 72℃ for 4 m, and a final extension of 5 m at 72℃. Groups of 8 reactions were pooled to yield ~250 μL and purified with Ampure XP Beads (0.6x ratio; Beckman Coulter), washed twice with 75% Ethanol, and eluted in 40 μL nuclease-free water. The DNA concentration of each sample was determined using Qubit High-Sensitivity DNA kit (Thermo Fisher). To prepare the Illumina sequencing library, 600 pg cDNA from each pooled sample was used in a modified Nextera XT library preparation (Illumina) (*Soumillon et al., 2014*) using the P5NEXTPT5 primer and extending the tagmentation time to 15 m. The resulting libraries were purified according to the Nextera XT protocol (0.6x ratio) and quantified by qPCR using Kapa Library Quantification (Kapa Biosystems). Six to eight sequencing libraries were loaded on a NextSeq High Output flow cell reading 26 bases in Read 1, including the spacer, sample barcode and UMI, eight bases in the i7 index read, and 50 bases in Read two representing the cDNA fragment from the 3' end of the transcript. Fastq files were pre-filtered to select reads (and their mates) that contained a cell barcode. Extracted reads were concatenated into a single pair of fastqs for downstream analysis. Sequencing adapters were trimmed and aligned as done for the bulk RNA-seq data. Gene counts were generated with the STARsolo algorithm; detailed parameters are given in the GEO submission file metadata.

## Connectomics analysis

The primary data used for our analyses are described in *Zheng et al., 2018* for the FAFB dataset and *Scheffer et al., 2020* for the hemibrain dataset. Hemibrain data was queried using NeuPrint and v1.1 of the connectome (neuprint.janelia.org). The unique identifier (bodyID number in the hemibrain v1.1 database) for neurons shown in figures, as well as additional data on other neurons connected to aIPg and pC1 neurons, are shown in *Supplementary file 2*. Visualizations of neuronal morphologies from the hemibrain dataset were generated in NeuTu (*Zhao et al., 2018*). Cytoscape (cytoscape.org) was used to produce the node layout of connectivity diagrams of connections between neurons, which were then edited in Adobe Illustrator. Thresholds were used in order to limit the number of neurons in the figures to those connections with the most synapses. In all cases, a threshold of three synapses was applied to connections between individual cells. Higher specific

thresholds, when applied, are specified in each figure legend. In *Figures 6*, *8* and *11*, a few neurons of interest that fell below the general thresholds used were also included. BodyIDs for neurons in connectomics figures are included in *Supplementary files 2* and *3*. A complete list of synaptic connections can be found in NeuPrint.

Anatomical videos were produced using Blender (blender.org) and software written by *Hubbard, 2020*. Narration was recorded using Camtasia (techsmith.com) and text, diagrams and narration were added to videos using Adobe Premiere Pro.

## Statistics

No statistical methods were used to pre-determine sample size. Sample size was based on previous literature in the field and experimenters were not blinded in most conditions as almost all data acquisition and analysis were automated. However, inactivation experiments in which manual quantification was performed were blinded. Biological replicates completed at separate times using different parental crosses were performed for each of the behavioral experiments. Behavioral data are representative of at least two independent biological repeats, only one of which is typically shown. For figures in which the behavioral data over the course of a trial is shown, pink shading indicates the stimulus period, the mean is represented as a solid line, and shaded error bars represent variation between experiments.

For each experiment, the experimental and control flies were collected, treated and tested at the same time. A Mann-Whitney *U* test or Kruskal-Wallis test and Dunn's post hoc test was used for statistical analysis. All statistical analysis was performed using Prism Software (GraphPad, version 7). p values are indicated as follows: ****$p<0.0001$; ***$p<0.001$; **$p<0.01$; and *$p<0.05$. See *Supplementary file 3* for exact *p*-values for each figure.

Boxplots show median and interquartile range (IQR). Lower and upper whiskers represent $1.5 \times$ IQR of the lower and upper quartiles, respectively; boxes indicate lower quartile, median, and upper quartile, from bottom to top. When all points are shown, whiskers represent range and boxes indicate lower quartile, median, and upper quartile, from bottom to top. Shaded error bars on graphs are presented as mean ± s.e.m.

## Acknowledgements

We thank D J Anderson and V Chiu (Caltech), V Ruta (Rockefeller), and the Janelia community, in particular: G Card, E Gruntman, U Heberlein, K Longden, A Nern, M Reiser, N Spruston, K Wang, F Wang, and T Wolff for their helpful suggestions during the course of this work and their comments on the manuscript. We thank the Janelia Project Technical Resources team (led by Gudrun Ihrke) for assistance with the manual correction of automated fly trajectories (Rebecca Arruda and NC) and performing neuronal segmentation of confocal images (Claire Managan). We also thank the Quantitative Genomics facility at Janelia for performing RNA profiling of the aIPg, pC1d, and pC1e lines, M Eddison and J Simon for their help in setting up the inactivation experiments and Igor Siwanowicz and Wyatt Korff for their assistance with the high-speed videos. We thank the Fly Light team for generating the images of GAL4 expression patterns and the Connectome Annotation Team for their help with proofreading. We also thank Katharina Eichler, Alia Suleiman, Bryon Eubanks, Nicholas Padilla, and Paul K LaFosse for their annotation of the FAFB dataset and Emily Joyce for video narrations. We thank Feng Li for help with connectomics analysis and producing the images shown in *Figure 10—figure supplement 2*. We also thank M Murthy and D Deutsch (Princeton) for exchanging information prior to publication.

## Additional information

### Funding

| Funder | Author |
| --- | --- |
| Howard Hughes Medical Institute | Catherine E Schretter<br>Yoshi Aso<br>Alice A Robie<br>Marisa Dreher |

Michael-John Dolan
Nan Chen
Masayoshi Ito
Tansy Yang
Ruchi Parekh
Kristin M Branson

The funders had no role in study design, data collection and interpretation, or the decision to submit the work for publication.

## Author contributions

Catherine E Schretter, Conceptualization, Data curation, Formal analysis, Supervision, Validation, Investigation, Visualization, Methodology, Writing - original draft, Project administration, Writing - review and editing; Yoshinori Aso, Conceptualization, Resources, Data curation, Supervision, Validation, Investigation, Writing - review and editing; Alice A Robie, Software, Formal analysis, Methodology, Writing - review and editing; Marisa Dreher, Formal analysis, Validation, Investigation, Visualization; Michael-John Dolan, Resources, Investigation; Nan Chen, Investigation; Masayoshi Ito, Resources; Tansy Yang, Validation, Investigation; Ruchi Parekh, Project administration; Kristin M Branson, Software, Formal analysis, Funding acquisition, Methodology, Writing - review and editing; Gerald M Rubin, Conceptualization, Resources, Data curation, Supervision, Funding acquisition, Validation, Investigation, Visualization, Writing - original draft, Project administration, Writing - review and editing

## Author ORCIDs

Catherine E Schretter (iD) https://orcid.org/0000-0002-3957-6838
Yoshinori Aso (iD) https://orcid.org/0000-0002-2939-1688
Marisa Dreher (iD) http://orcid.org/0000-0002-0041-9229
Michael-John Dolan (iD) http://orcid.org/0000-0001-9666-3682
Tansy Yang (iD) http://orcid.org/0000-0003-1131-0410
Ruchi Parekh (iD) http://orcid.org/0000-0002-8060-2807
Gerald M Rubin (iD) https://orcid.org/0000-0001-8762-8703

## Decision letter and Author response

Decision letter https://doi.org/10.7554/eLife.58942.sa1
Author response https://doi.org/10.7554/eLife.58942.sa2

# Additional files

## Supplementary files

• Supplementary file 1. Framewise performance of automated classifiers.

• Supplementary file 2. List of hemibrain v1.1 unique neuron identifier numbers and synapse numbers for connected neurons.

• Supplementary file 3. Sample size and statistics for behavioral analysis.

• Transparent reporting form

## Data availability

Data is included in the paper and the supplementary files. Source data have been provided for Figures 1,2, 4 and 9.

The following dataset was generated:

| Author(s) | Year | Dataset title | Dataset URL | Database and Identifier |
|---|---|---|---|---|
| Schretter CE, Aso Y, Robie AA, Dreher M, Dolan MJ, Chen N, Ito M, | 2020 | Cell types and neuronal circuitry underlying female aggression in *Drosophila* | http://www.ncbi.nlm.nih.gov/geo/query/acc.cgi?acc=GSE158748 | NCBI Gene Expression Omnibus, GSE158748 |

Yang T, Parekh R,
Branson K, Rubin
GM

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
