## [Decision Letter]

**Acceptance summary:**

This manuscript by Schretter et al. adds to the growing understanding of neural circuits controlling female social behaviors of *Drosophila*. It identifies two types of sexually dimorphic neurons that regulate female aggressive interactions, demonstrates their function using optogenetics and, by tracing their connections using the fly EM connectome, shows recurrent connections between specific subsets of these two neuronal classes. This comprehensive description of the synaptic inputs and outputs of aIPg and pC1d neurons using the EM-based circuit mapping, thoughtfully described and discussed, identifies new connected elements of female aggression. It further leads to the definition of pC1d neurons as a central hub receiving inputs from several neurons (oviN and pC1a) implicated in female behaviors such as oviposition (Wang et al., 2020), as well as feedback from aIPg neurons.

**Decision letter after peer review:**

Thank you for submitting your article "Neuronal circuitry underlying female aggression in *Drosophila*" for consideration by *eLife*. Your article has been reviewed by three peer reviewers, and the evaluation has been overseen by a Reviewing Editor and Catherine Dulac as the Senior Editor. The reviewers have opted to remain anonymous.

The reviewers have discussed the reviews with one another and the Reviewing Editor has drafted this decision to help you prepare a revised submission.

As the editors have judged that your manuscript is of interest, but as described below that additional experiments are required before it is published, we would like to draw your attention to changes in our revision policy that we have made in response to COVID-19 (https://elifesciences.org/articles/57162). First, because many researchers have temporarily lost access to the labs, we will give authors as much time as they need to submit revised manuscripts. We are also offering, if you choose, to post the manuscript to bioRxiv (if it is not already there) along with this decision letter and a formal designation that the manuscript is "in revision at *eLife*". Please let us know if you would like to pursue this option.

While we appreciate that this may not be attractive or easily possible, another option that you and your co-authors may wish to consider is to submit a revised manuscript addressing at least points 1-5, as an *eLife* Short Report, compressed and refocused on the main discovery of two new connected circuit elements involved in female aggression.

As this decision was being finalised, we were alerted to an overlapping manuscript by Mala Murthy and colleagues (Deutsch et al., 2020), which is being independently reviewed by *eLife*. As there is no reason for both labs to independently perform the same sets of experiments, it will be acceptable for a revised manuscript to refer to confirmatory or additional observations made by Murthy and colleagues if these are useful to strengthen or establish conclusions reached here.

Summary:

This manuscript by Schretter et al. adds to the growing understanding of neural circuits controlling female social behaviors of *Drosophila*. It identifies two types of sexually dimorphic neurons that regulate female aggressive interactions, demonstrates their function using optogenetics and, by tracing their connections using the fly EM connectome, shows recurrent connections between specific subsets of these two neuronal classes.

Optogenetic activation of aIPg neurons promotes aggression toward both male and female targets. Interestingly, brief aIPg neuron activation results in persistently increased aggression during light offset period, suggesting a role for aIPg activation in driving persistent female aggressive behavior. Moreover, inhibiting synaptic transmission in aIPg neurons attenuates innate aggression, indicating a physiological requirement that these neurons promote aggression in female flies. The authors further demonstrate that optogenetic activation of the previously characterized doublesex-expressing pC1 neurons also result in elevated levels of aggression. Using of split GAL4 drivers labelling specific subtypes of pC1 neurons, authors specify pC1d, as the one of five pC1 cell types that drive female aggressive behaviors. Using the EM connectome to trace connections of these cells, the authors discovered that pC1d and pC1e (but not other pC1 cell types) are presynaptic to aIPg neurons. While pC1d is the major pre-synaptic input to all three aIPg neuron subtypes (aIPg type 1 aIPg type 2, and aIPg type 3) identified in by EM, only one, aIPg type1, provides strong reciprocal feedback input to pC1d. These reciprocal connections may explain increased aggression extended beyond the activation period in flies expressing CsChrimson in aIPg neurons.

This comprehensive description of the synaptic inputs and outputs of aIPg and pC1d neurons using the EM-based circuit mapping, thoughtfully described and discussed, identifies new connected elements of female aggression. It further leads to the definition of pC1d neurons as a central hub receiving inputs from several neurons (oviN and pC1a) implicated in female behaviors such as oviposition (Wang et al., 2020), as well as feedback from aIPg neurons.

Essential revisions:

There are two main lines of concern. One line of concern that needs to be addressed, requires relatively straightforward experiments (Essential revisions 1-5 below) to better establish the primary observations made and conclusions reached in this study. The second, major line of concern regards the relatively descriptive nature of the circuit mapping, which leaves issues of functional connectivity between the neuronal classes as well as their biological significance lightly established and difficult to appreciate. It is likely that experiments to address these issues (Essential revisions 6-7) will require new reagents that are not currently available. In light of this, the reviewers agree that though useful, these will not be necessary if the authors choose to submit a revised manuscript as an appropriately abbreviated, refocussed and retitled *eLife* Short Report.

1) The conclusion of a "persistent" effect of aIPg neuron activation on female aggression needs to be supported by an additional experiment. In addition to a relatively modest level of aggression after the stimulus, the experiments in Figure 1 and the rest of the paper do not exclude a possibility that it is the aggressive interaction itself that provides a momentum to persists after the stimulus is turned off. An experiment similar to the one in Hoopfer et al., 2015 (where flies were allowed to interact after the stimulus was turned off) would be necessary to demonstrate that the "persistence" is due to the activity of neural circuit, rather than residual effects of intense aggressive interactions. The authors admission that the ethological relevance of persistence is unknown, and suggestion that the persistence may be "an experimental artefact" may be indicative of a some doubt on part of the authors.

Even if the additional behavioral experiments supports their conclusion on aIPg neurons, the additional conclusion that the "recurrent connectivity" between pC1d and aIPg is important for behavioral "persistence" will need to be either modified convincingly demonstrated (See points 6-7).

2) In Figure 3, the authors show that activation of aIPg neurons promote aggression toward both male and female targets. Does the same manipulation induce aggressive behaviors in a solitary female? If not, why is this manipulation insufficient to cause the phenotype?

3) In Figure 3, "time performing aggressive behaviors" looks smaller toward a wild-type male target than toward a female target. How do they explain this result? Is the decreased aggression caused by male- pheromones which suppress female aggression?

4) From the results shown in Figure 3—figure supplement 2, the authors conclude that the copulation latency is not affected by optogenetic stimulation of aIPg neurons, but the sample size looks small in these experiments. Could authors increase the sample size and see whether their conclusions hold? Particularly since activation of aIPg neurons causes aggression towards males, one might expect copulation to be impacted in these females?

5) It often is not clear how many times each experiment was performed. For example – data in Figure 2 – appear to be derived from data pooled from two experiments done during the same week. Were these animals from the same cross? How many times was this done independently? Similar, detailed information should be provided for each of the figures. A total number of flies is presented each time, but it is usually not clear how many independent samples there are.

6) The authors emphasize the significance of recurrent connectivity between pC1d and aIPg neurons as the basis for female aggressive behavior. The recurrent connectivity is observed specifically between pC1d and aIPg Type 1 neurons, but no data demonstrates that it is the Type 1 neurons that is responsible for female aggression. It is equally possible that Type 2 or 3 is the important subset. These 2 types do not make recurrent connection with pC1d – instead, these are almost unidirectional downstream of pC1d (Figure 8A) (note that Type 2 is the second most prominent downstream of pC1d, as shown in Figure 10C). The authors did not perform any physiological experiments to demonstrate whether these recurrent (or unidirectional) connectivities are functional. Without evidence on which aIPg subtype is important for female aggression, it would be fair to acknowledge that top-down connection from pC1d to aIPg could be the behaviorally relevant functional motif between the 2 neuronal populations.

7) The recurrent model is difficult to reconcile with the different behavioral phenotypes caused by the activation of pC1d and aIPg neurons, as (at least qualitatively) these two reciprocally connected neurons would have similar effect on the circuit. For example, why are aIPg neurons necessary for female aggression (Figure 2), but not pC1d (Figure 5—figure supplement 10), if these make reciprocal connection? If the connectivity is uni-directional, however, it is possible to speculate that aIPg neurons are necessary to integrate important information, one of which (not necessary for aggression on its own) is provided by pC1d. Ultimately, it would be necessary to perform "epistasis" experiments between the 2 populations (e.g. silencing one while activating the other) to resolve which of the 2 motifs is more relevant, although this would be more than what can be reasonably expected in a normal period allowed for revisions.

---

## [Author Response]

[…] As this decision was being finalised, we were alerted to an overlapping manuscript by Mala Murthy and colleagues (Deutsch et al., 2020), which is being independently reviewed by eLife. As there is no reason for both labs to independently perform the same sets of experiments, it will be acceptable for a revised manuscript to refer to confirmatory or additional observations made by Murthy and colleagues if these are useful to strengthen or establish conclusions reached here.

We are grateful for the suggestions and have followed the recommendation in producing our revised manuscript. In particular, Deutsch et al., 2020, focuses heavily on the mechanism of persistence of female aggression as their paper is entitled: “The Neural Basis for a Persistent Internal State in *Drosophila* Females”. We therefore largely eliminated our speculation about the mechanism of persistence and referred to their experimental data as an alternative to doing more mechanistic experiments ourselves. To reflect the fact that, as noted in the review summary, we discovered a new cell type (aIPg) and determined which member of the pC1 cluster was critical (pC1d), we modified our title from “Neuronal circuitry underlying female aggression in *Drosophila*” to “Cell types and neuronal circuitry underlying female aggression in *Drosophila*.”

Before describing in detail how we answered each of the reviewer’s comments, we would also like to point out one area of enhancement we made in our paper, concerning changes to visual behaviors after aIPg activation. More specifically, by performing assays in a darkened arena, we had shown that aggression was reduced but did not explore this observation in detail in the original manuscript. In the revised manuscript, we report the results of additional computer vision analysis of this data to determine metrics such as bout duration when flies were deprived of visual input. We also moved their presentation from a figure supplement (Figure 1—figure supplement 8) to a main figure (revised manuscript Figure 9) and added a video (revised manuscript Video 15).

Two additional new findings published since our initial submission prompted us look more closely at connectomic information that might provide clues as to how aIPg activation could influence the processing of visual information. First, a paper from Rachel Wilson’s lab (Rayshubskiy et al., 2020) showed that a particular descending neuron cell type, DNa02, played an important role in steering of the fly’s movements. We had identified DNa02 as a downstream target of two of aIPg’s direct targets (original Figure 11) but did not appreciate its behavioral significance until their manuscript. Secondly, a paper from Vanessa Ruta’s laboratory on male behavior (Sten et al., 2020), showed that arousal in males gated visual input from LC10 visual neurons, a population of lobular columnar neurons. This is the same population of visual projection neurons that our lab (Wu et al., 2016) had previously shown to trigger reaching and approach. Additionally, the Dickson and Borst labs had shown this neuronal population was important for the male to track the female in courtship (Riberio et al., 2018). Strikingly, we found that LC10 and aIPg both provide input to eight different interneuron cell types that connect the anterior optical tubercule to seven different descending neuron cell types, including DNa02. We added figures diagramming these connections (Figure 10, Figure 10—figure supplements 1 and 2) as well as a model figure (Figure 12) to present these new observations. We are excited about these observations because they suggest specific circuit mechanism for how aIPg-induced changes in “brain state” might modify behavior. While these are new additions to our paper, we wish to emphasize that they did not involve new experimental work but are simply the result of more extensive analysis of the recently published (Scheffer et al., 2020) fly connectome. We believe their inclusion will greatly increase the impact and general interest of our paper.

Essential revisions:There are two main lines of concern. One line of concern that needs to be addressed, requires relatively straightforward experiments (Essential revisions 1-5 below) to better establish the primary observations made and conclusions reached in this study. The second, major line of concern regards the relatively descriptive nature of the circuit mapping, which leaves issues of functional connectivity between the neuronal classes as well as their biological significance lightly established and difficult to appreciate. It is likely that experiments to address these issues (Essential revisions 6-7) will require new reagents that are not currently available. In light of this, the reviewers agree that though useful, these will not be necessary if the authors choose to submit a revised manuscript as an appropriately abbreviated, refocussed and retitled eLife Short Report.1) The conclusion of a "persistent" effect of aIPg neuron activation on female aggression needs to be supported by an additional experiment. In addition to a relatively modest level of aggression after the stimulus, the experiments in Figure 1 and the rest of the paper do not exclude a possibility that it is the aggressive interaction itself that provides a momentum to persists after the stimulus is turned off. An experiment similar to the one in Hoopfer et al., 2015 (where flies were allowed to interact after the stimulus was turned off) would be necessary to demonstrate that the "persistence" is due to the activity of neural circuit, rather than residual effects of intense aggressive interactions. The authors admission that the ethological relevance of persistence is unknown, and suggestion that the persistence may be "an experimental artefact" may be indicative of a some doubt on part of the authors.Even if the additional behavioral experiments supports their conclusion on aIPg neurons, the additional conclusion that the "recurrent connectivity" between pC1d and aIPg is important for behavioral "persistence" will need to be either modified convincingly demonstrated (See points 6-7).

As mentioned above, we removed speculation about the possible connection between recurrent connectivity and persistence from our paper. We did not alter the data previously presented as these were not in question, but we did remove the speculation about possible underlying mechanism and instead refer to Deutsch et al., 2020, throughout as suggested by the *eLife* editors.

2) In Figure 3, the authors show that activation of aIPg neurons promote aggression toward both male and female targets. Does the same manipulation induce aggressive behaviors in a solitary female? If not, why is this manipulation insufficient to cause the phenotype?

We thank the reviewers for this suggestion and have added the requested data examining changes in the behavior of an isolated females when aIPg neurons are activated. We found no change in the velocity of solitary females before and after activation (revised manuscript Figure 2—figure supplement 2; subsection “Activation of aIPg overrides the requirement for specific environmental conditions for female aggressive behaviors”), but most of our verified classifiers assess social behaviors (chasing, touching, aggression) and were not applicable. As it is unclear what changes one might expect, we used careful human observation of these videos and could discern no obvious changes in behavior. We added a new video (revised manuscript Video 5) showing four examples each of EmptySS>Chrimson control and aIPg>Chrimson individual flies following activation to allow the interested reader to examine typical primary data.

The reviewers ask why such a manipulation might be insufficient to cause a phenotype in an isolated fly. Prompted by this question, we added text in the Discussion (subsection “Activation of aIPg neurons results in a change to the individual’s state”) explaining that we do not consider aIPg to be a command neuron, but rather that aIPg activation influences the propensity to engage in certain social behaviors. In that view, the lack of change in the behavior of an individual fly is not unexpected.

3) In Figure 3, "time performing aggressive behaviors" looks smaller toward a wild-type male target than toward a female target. How do they explain this result? Is the decreased aggression caused by male- pheromones which suppress female aggression?

We performed this experiment using two different split-GAL4 lines that label aIPg neurons (original Figure 3 and Figure 3—figure supplement 1; revised manuscript Figure 2 and Figure 2—figure supplement 1). In one case, activated females exhibited slightly less aggression towards a wild-type male than a wild-type female target fly, while the other aIPg line showed slightly more aggression towards a wild-type male target fly. Neither change was significant (*p*=0.65 and *p*=0.16, respectively), which is now noted in the figure legends. We conclude that there is no consistent difference.

Due to the variability in the data noted by the reviewers, we have also added the *p*-values for a representative biological repeat of this data to the figure legend for Figure 2. We also note that we make no claim in the paper about differences between the level of aggression to targets of different sexes. We only claim that there is also aggression toward male targets, which both our lines show at highly significant levels (*p*=0.01 or *p*=0.001, respectively).

4) From the results shown in Figure 3—figure supplement 2, the authors conclude that the copulation latency is not affected by optogenetic stimulation of aIPg neurons, but the sample size looks small in these experiments. Could authors increase the sample size and see whether their conclusions hold? Particularly since activation of aIPg neurons causes aggression towards males, one might expect copulation to be impacted in these females?

We thank the reviewers for this suggestion and have doubled the number of flies used in this experiment (Figure 2—figure supplement 3 in the revised manuscript). With the larger sample size, we also found the same conclusion. Additionally, while there was no significant difference at either sample size, the activated and control flies had more similar latency times after the number of flies used in the experiment increased (original means of EmptySS>Chrimson: 101.4±28.2 vs. aIPgSS1>Chrimson: 170.4±40.8 and with larger n EmptySS>Chrimson: 253.4±44.51 vs. aIPgSS1>Chrimson: 238.3±40.02)

5) It often is not clear how many times each experiment was performed. For example – data in Figure 2 – appear to be derived from data pooled from two experiments done during the same week. Were these animals from the same cross? How many times was this done independently? Similar, detailed information should be provided for each of the figures. A total number of flies is presented each time, but it is usually not clear how many independent samples there are.

We apologize for this confusion. All the requested details were contained in a supplementary file in the original submission. We have now moved this information to a move obvious location in the figure legends and Materials and methods section of the main manuscript.

6) The authors emphasize the significance of recurrent connectivity between pC1d and aIPg neurons as the basis for female aggressive behavior. The recurrent connectivity is observed specifically between pC1d and aIPg Type 1 neurons, but no data demonstrates that it is the Type 1 neurons that is responsible for female aggression. It is equally possible that Type 2 or 3 is the important subset. These 2 types do not make recurrent connection with pC1d – instead, these are almost unidirectional downstream of pC1d (Figure 8A) (note that Type 2 is the second most prominent downstream of pC1d, as shown in Figure 10C). The authors did not perform any physiological experiments to demonstrate whether these recurrent (or unidirectional) connectivities are functional. Without evidence on which aIPg subtype is important for female aggression, it would be fair to acknowledge that top-down connection from pC1d to aIPg could be the behaviorally relevant functional motif between the 2 neuronal populations.

As with point #1, we address this question mainly by referring the reader to the paired Deutsch et al., 2020 paper for a discussion of recurrent connectivity. We certainly did not mean to imply in our original manuscript that we had determined the mechanism of persistence; rather we simply intended to point out circuit motifs that were well positioned to play such a role. This should now be clear in the revised manuscript, where we deleted nearly all of that speculation. In addition, we added text to the revised paper that explicitly states that the inputs and outputs of the different aIPg types differ and that they are differ greatly from those of pC1d (subsection “aIPg and pC1d neurons are interconnected, but have largely distinct upstream and downstream partners”). We also make it clear to the reader that all our behavioral data was done with split-GAL4 lines that include aIPg types 1, 2 and 3. Therefore, we cannot assign behavioral roles to individual types based on separately activating the individual types.

7) The recurrent model is difficult to reconcile with the different behavioral phenotypes caused by the activation of pC1d and aIPg neurons, as (at least qualitatively) these two reciprocally connected neurons would have similar effect on the circuit. For example, why are aIPg neurons necessary for female aggression (Figure 2), but not pC1d (Figure 5—figure supplement 10), if these make reciprocal connection? If the connectivity is uni-directional, however, it is possible to speculate that aIPg neurons are necessary to integrate important information, one of which (not necessary for aggression on its own) is provided by pC1d. Ultimately, it would be necessary to perform "epistasis" experiments between the 2 populations (e.g. silencing one while activating the other) to resolve which of the 2 motifs is more relevant, although this would be more than what can be reasonably expected in a normal period allowed for revisions.

We made some writing changes to clarify the point raised by the reviewer. More specifically we point out the extensive differences between the upstream inputs and downstream targets of aIPg and pC1d neurons (subsection “aIPg and pC1d neurons are interconnected, but have largely distinct upstream and downstream partners”), making it clear why we would not expect identical phenotypes from their activation. This would be irrespective of the possible role of reciprocal connections, as these provide only a minority of inputs to either pC1d or aIPg cell types.

Both pC1d and aIPg cell types induce aggression, so in that sense they do have qualitatively similar behavioral effects. We do believe [as suggested by the reviewers: “it is possible to speculate that aIPg neurons are necessary to integrate important information, one of which (not necessary for aggression on its own) is provided by pC1d.”] that the simplest interpretation of our data is that pC1d acts upstream of aIPg, and the phenotypes seen with inactivation aIPg are consistent with that view. We also think that the aIPg and pC1d neurons are likely to act in parallel to produce different aspects of the phenotype. We have modified the writing to clarify these points. Our new observation of aIPg’s connections that are well-positioned to gate visual behavior also support this view. We agree with the reviewer that formal epistasis experiments might be informative, but these require reagents that do not yet exist. In any case, as mentioned in our response to point 6, we have removed speculation about the potential role of the reciprocal connections as we agree it will require additional experiments to determine what role, if any, they play.